



# SoilGrids 2.0: producing quality-assessed soil information for the globe

Luis M. de Sousa[1], Laura Poggio[1], Niels H. Batjes[1], Gerard B. M. Heuvelink[1], Bas Kempen[1], Eloi Ribeiro[1], and David Rossiter[1]

[1]ISRIC - World Soil Information - Wageningen (NL)

**Correspondence:** Laura Poggio (laura.poggio@isric.org)

**Abstract.** SoilGrids produces maps of soil properties for the entire globe at medium spatial resolution (250 metres cell size) using state-of-the-art machine learning methods to generate the necessary models. It takes as inputs soil observations from about 240 000 locations worldwide and over 400 global environmental covariates describing vegetation, terrain morphology, climate, geology and hydrology. The aim of this work was the production of quality-assessed global maps of soil properties, with cross-validation, hyper-parameters selection and quantification of spatially explicit uncertainty, as implemented in the SoilGrids version 2.0 product incorporating state of the art practices and adapting them for global digital soil mapping with legacy data. The paper presents the evaluation of the global predictions produced for soil organic carbon content, total nitrogen, coarse fragments, pH(water), cation exchange capacity, bulk density and texture fractions at six standard depths (up to 200 cm). The quantitative evaluation showed metrics in line with previous global, continental and large regions studies. The qualitative evaluation showed that coarse scale patterns are well reproduced. The spatial uncertainty at global scale highlighted the need for more soil observations, especially in high latitude regions.

## 1 Introduction

Healthy soils provide important ecosystem services at the local, landscape and global level, and are important for the functioning of terrestrial ecosystems (Banwart et al., 2014; FAO and ITPS, 2015; UNEP, 2012). Up to date information on world soil resources, at a scale level commensurate with user needs, is required to address a range of pressing global issues. These include avoiding and reducing soil erosion through land rehabilitation and development (Borrelli et al., 2017; WOCAT, 2007), mitigation and adaptation to climate change (Batjes, 2019; Harden et al., 2017; Sanderman et al., 2017; Yigini and Panagos, 2016; Smith et al., 2019), ensuring water security (Rockstroem et al., 2012), food production and food security (FAO et al., 2018; Soussana et al., 2017; Springmann et al., 2018), as well as the preservation of biodiversity (Barnes, 2015; IPBES, 2019; van der Esch et al., 2017) and human livelihood (Bouma, 2015).

Quality-assessed soil data are required to support the Land Degradation Neutrality (LDN) (Cowie et al., 2018) initiative, achieve several of the Sustainable Development Goals, and provide input for e.g. earth system modelling by the IPCC (Dai et al., 2019; Luo et al., 2016; Todd-Brown et al., 2013) and crop modelling (Han et al., 2019; van Bussel et al., 2015; van Ittersum et al., 2013), among many other applications. Such information can in turn help inform international conventions





such as the United Nations Framework Convention on Climate Change (UNFCCC), the United Nation Convention to Combat Desertification (UNCCD), and the United Nations Convention on Biological Diversity (UNCBD).

Until the last decade, most global scale assessments requiring soil data used the Digital Soil Map of the World (DSMW) FAO (1995), an updated version of the original printed 1:5 M scale Soil Map of the World (SMW) (FAO-Unesco, 1971-1981). The soil-geographical data from the DSMW provided the basis for generating a range of derived soil property databases that drew on a larger selection of soil profile data held in the WISE database (Batjes, 2012) and more sophisticated (taxotransfer) procedures for deriving various soil properties (Batjes et al., 2007). Subsequently, in a joint effort coordinated by the Food and Agriculture Organization of the United Nations (FAO), the best available (newer) soil information collated for central and southern Africa, China, Europe, northern Eurasia, and Latin America were combined into a new product known as the Harmonised World Soil Database (HWSD) (FAO et al., 2012).

Until recently, the HWSD was the only digital map annex database available for global analyses. However, it has several limitations (GSP and FAO, 2016; Hengl et al., 2014; Ivushkin et al., 2019; Omuto et al., 2012). Some of these relate to the partly outdated soil-geographic data as well as the use of a two-layer model (0-–30 and 30-–100 cm) for deriving soil properties. Others concern the derived attribute data themselves, in particular their unquantified uncertainty, and the use of three different versions of the FAO legend (i.e. FAO74, FAO85 and FAO90). These issues have been addressed to varying degrees in various new global soil datasets (Batjes, 2016; Shangguan et al., 2014; Stoorvogel et al., 2017) that still largely draw on a traditional, pedology-based mapping approach (Dai et al., 2019).

The last decade, Digital Soil Mapping (DSM) became a widely used approach to obtain maps of soil information (Minasny and McBratney, 2016). DSM consists primarily in building a qualitative numerical model between soil observations and environmental information acting as proxies for the soil forming factors (McBratney et al., 2003; Minasny and McBratney, 2016). The number of studies using DSM to produce maps of soil properties is ever growing. Numerous modelling approaches are considered, from linear model to geo-statistics, machine learning and artificial intelligence (e.g. deep learning). Keskin and Grunwald (2018) provide a recent review of methods and applications in the field of DSM. DSM techniques have been applied at various spatial resolutions (e.g. 30m to 1000m) to support precision farming (Piikki et al., 2017) as well as applications at landscape (Ellili et al., 2019; Kempen et al., 2015), country (Mora-Vallejo et al., 2008; Nijbroek et al., 2018; Vitharana et al., 2019; Poggio and Gimona, 2017b; Kempen et al., 2019), regional (Dorji et al., 2014; Moulatlet et al., 2017), continental (Grunwald et al., 2011; Guevara et al., 2018; Hengl et al., 2017a), and global level (Hengl et al., 2014, 2017b; GSP and ITPS, 2018; Stockmann et al., 2015).

The aim of this paper is to present the development of new soil property maps for the world at 250 metres grid resolution with a process incorporating state-of-the-art practices and adapting them to the challenges of global Digital Soil Mapping with legacy data. It builds on previous global soil properties maps (SoilGrids250m) (Hengl et al., 2017b), integrating up-to-date machine learning methods, the increased availability of standardised soil profile data for the world (Batjes et al., 2020) and environmental co-variates (Nussbaum et al., 2018; Poggio et al., 2013; Reuter and Hengl, 2012). In particular, this paper addresses at global scale the following elements:





1. quality-assessed soil profile data derived from ISRIC's World Soil Information Service (WoSIS), with expanded number and spatial distribution of observations (Batjes et al., 2020);

2. a reproducible co–variate selection procedure, relying on Recursive Feature Elimination (Guyon et al., 2002);

3. improved cross-validation procedure, based on spatial stratification; and

4. quantification of prediction uncertainty using Quantile Regression Forest (Meinshausen, 2006).

## 2 Materials and Methods

This study uses Quantile Regression Forest (Meinshausen, 2006), a method with a limited number of parameters to be tuned and that has proven an effective compromise between accuracy and feasibility for large datasets. Selected primary soil properties as defined and described in the GlobalSoilMap specifications (Arrouays et al., 2014) were modelled. The following sections describe each step of the workflow (Figure 1) in detail. These include:

1. Input soil data preparation

2. Covariates selection

3. Model tuning and cross-validation

4. Final model fitting for prediction

5. Predictions with uncertainty estimation.

### 2.1 Soil observation data

Soil property data for this study were derived from the ISRIC World Soil Information Service (WoSIS), which provides consistent, standardised soil profile data for the world (Batjes et al., 2020). All soil data shared with ISRIC to support global mapping activities are first stored in the ISRIC Data Repository together with their metadata (including the name of the data owner and licence defining access rights). Subsequently, the source data are imported 'as is' into PostgreSQL, after which they are ingested into the WoSIS data model itself. Following data quality assessment and control, the descriptions for the soil analytical methods and the units of measurement are standardised using consistent procedures (Ribeiro et al., 2018). Ultimately, upon final consistency checks, the quality-assessed and standardised data are made available via the ISRIC Soil Data Hub (https://data.isric.org) in accord with the licence specified by the data providers. As a result, not all data standardized in WoSIS are freely available to the international community. Hence, this study considers two 'sources' of point data.

First, the latest publicly available snapshot of WoSIS (Batjes et al., 2020). It contains, among others, data for chemical (organic carbon, total nitrogen, soil pH, cation exchange capacity) and physical properties (soil texture (sand, silt, and clay), coarse fragments). The snapshot comprises 196 498 geo-referenced profiles originating from 173 countries, representing over





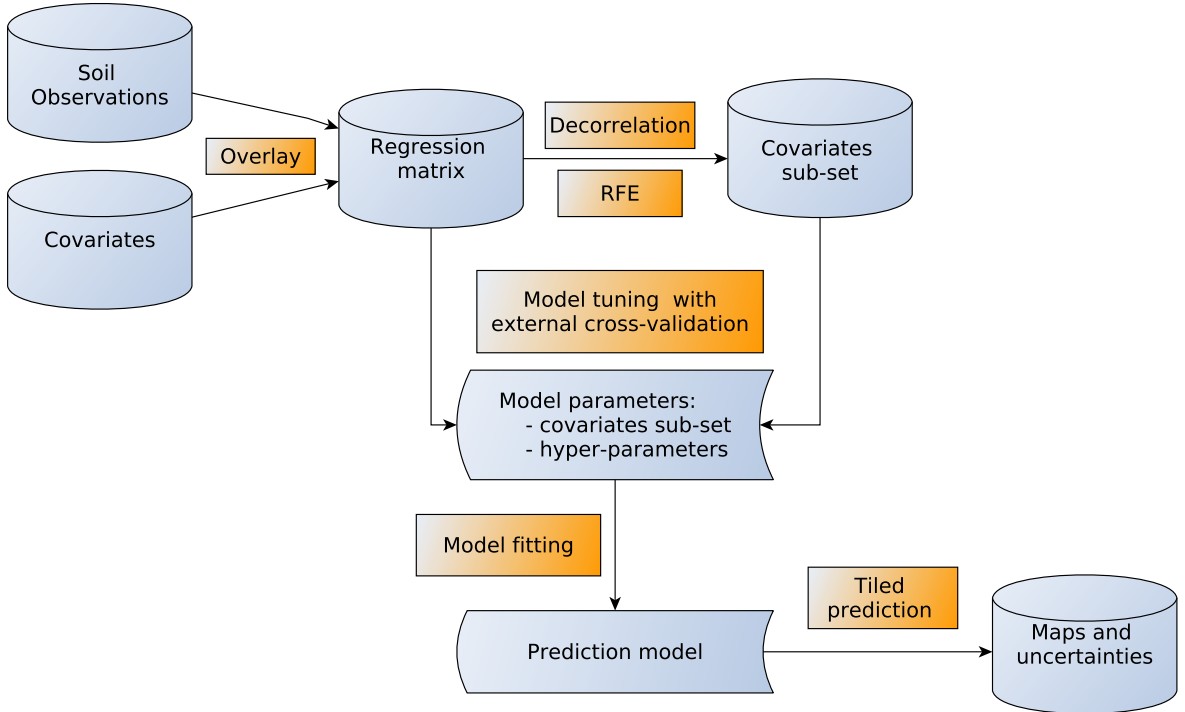

**Figure 1.** Workflow of the methodological approach.

832 000 soil layers (or horizons), in total over 5.8 million records. Generally, there are more observations for the superficial than the deeper layers. Detailed information about the snapshot can be found in Batjes et al. (2020).

Second, in addition to the freely-shareable data, several soil profile databases in our repository have licences stipulating that ISRIC may only use them for SoilGrids applications or visualizations, for example EU-LUCAS (Tóth et al., 2013) and soil data for the state of Victoria (Australia). The corresponding source data sets were screened and processed using the same procedures as used for the regular WoSIS workflow (some 42 000 profiles). As a result, some 240 000 profiles in total were used as the data source for the present 2020 SoilGrids run, comprising more than 920 000 observed soil layers. During data processing some minor corrections were made to the merged input dataset.

### 2.1.1 Soil properties

For the purposes of SoilGrids, "soil" is up to 2m thick unconsolidated material at the Earth's epidermis in direct contact with the atmosphere; thus sub-aqeous and tidally-exposed soils are not considered here, neither are materials deeper than 2m. This decision has consequences for computations of total stocks, in particular "soil" organic carbon.

Table 1 describes the soil properties that are considered in this version of SoilGrids: organic carbon content, total nitrogen content, soil pH (measured in water), cation exchange capacity, soil texture fractions, and proportion of coarse fragments.





**Table 1.** Soil properties description and units.

| Soil property | Acronym | Units | Mapped Units | Description |
|---|---|---|---|---|
| Bulk density | BDOD | kg/dm$^3$ | cg/cm$^3$ | Bulk density of the fine earth fraction oven dry |
| Cation exchange capacity | CEC | cmol(c)/kg | mmol(c)/kg | Capacity of the fine earth fraction to hold exchangeable cations |
| Coarse fragments | CFVO | cm$^3$/100cm$^3$ (volume %) | cm$^3$/dm$^3$ | Volumetric content of fragments larger than 2 mm in the whole soil |
| Nitrogen | N | g/kg | cg/kg | Sum of total nitrogen (ammonia, organic and reduced nitrogen) as measured by Kjeldahl digestion plus nitrate-nitrite |
| pH (water) | pH | - | ∗10 | Negative common logarithm of the activity of hydronium ions (H$^+$) in water |
| Organic carbon concentration | SOC | g/kg | dg/kg | Gravimetric content of organic carbon in the fine earth fraction of the soil |
| Soil texture fractions | STF | % | g/kg | Gravimetric contents of sand, silt and clay in the fine earth fraction of the soil |

These properties were modelled for the six standard depths intervals as defined in the GlobalSoilMap specifications (Arrouays et al., 2014): 0-5, 5-15, 15-30, 30-60, 60-100 and 100-200 cm.

'Litter layers' on top of minerals soils were excluded from further modelling using the following assumptions. Consistency in layer depth (e.g. sequential increase in the upper and lower depth reported for each layer down the profile) in WoSIS was checked using automated procedures. In accord with current internationally accepted conventions, such depth increments are given as "measured from the surface, including organic layers and mineral covers" (FAO, 2006; Schoeneberger et al., 2012). Prior to 1993, however, the start (zero depth) of the profile was set at the top of the mineral surface (the solum proper), except when "thick" organic layers as defined for peat soils (FAO-ISRIC, 1986) were present at the surface. Then the top of the peat layer was taken as the soil surface. Organic horizons were recorded as above and mineral horizons recorded as below, relative to the mineral surface (Schoeneberger et al., 2012) (pp. 2–6). Insofar as is possible, "superficial litter" on top of mineral layers was flagged as an auxiliary (Boolean) variable, also with reference to the original soil horizon designation when provided, so they can be filtered out during auxiliary computations of soil properties.

### 2.1.2 Transformation of texture data

A transformation was applied to the texture fractions, as follows. The relative percentage of sand, silt and clay can be treated as compositional variables, as the sum of the components always equals 100%. Therefore, these components were transformed





using the Addictive Log-Ratio (ALR) transformation with the Gauss–Hermite quadrature (Aitchison, 1986). ALR has previously been applied to soil texture data (Lark and Bishop, 2007; Akpa et al., 2014; Ballabio et al., 2016; Poggio and Gimona, 2017a), and it has been shown (Lark and Bishop, 2007) that ALR-transformed variables preserve information on the spatial correlation and maintain the compositional integrity of the original components. In this study, clay was used as the denominator

variable. Therefore the two ALR components that were interpolated can be defined as:

$$
\begin{aligned}
ALR1 &= log\left(\frac{\text{sand}}{\text{clay}}\right) \\
ALR2 &= log\left(\frac{\text{silt}}{\text{clay}}\right)
\end{aligned}
\tag{1}
$$

### 2.1.3 Spatial stratification of observations

Random splitting of profile observations into $n$ validation folds is not suitable in this context, considering the high spatial variation in observation density as it would provide biased results (Brus, 2014). For regions like Europe and North America

there are over 4 profiles per 10 km$^2$, whereas for large countries in Asia, such as Kazakhstan, India or Mongolia the number of available profiles is still quite limited (< 1 profile per 100 km$^2$) (see Batjes et al. (2020) for further details).

Therefore, soil observations were spatially stratified in the geodetic domain to guarantee a balanced spatial distribution within each validation fold. Spatial strata, in the form of hexagons, were created with an Icosahedral Snyder Equal-Area Grid (ISEAG) of aperture 3 and resolution 6, resulting in 7 292 strata, each with an area around 70 000 $km^2$. This ISEAG was

generated with the `dggridR` package for the R language (Barnes et al., 2016).

The profiles were assigned to one of ten folds, each equally represented in each stratum, i.e., each cell of the grid previously described. The `caret` R package was used. All observations (layers or horizons) belonging to a profile were always in the same fold for both model calibration and evaluation.

## 2.2 Environmental covariates

Over four hundred geographic layers were available as environmental covariates for this work. These were chosen for their presumed relation to the major soil forming factors, including long-term soil conditions, i.e., the "time" factor. Appendix A provides a list of the products used as covariates and their sources. The layers considered can be grouped as follow:

- **Climate**: temperature, precipitation, snowfall, cloud cover, solar radiation, wind speed.

- **Ecology**: bioclimatic zones and ecophysiographic regions.

- **Geology**: soil and sedimentary thickness, rock types.

- **Land Use/Cover**: from sources such as the European Space Agency (ESA) and U.S. Geological Survey (USGS).

- **Elevation and terrain morphology**: including numerous morphology indexes and landform classes.





- **Vegetation Indexes**: such as the Normalized Difference Vegetation Index (NDVI), Enhanced Vegetation Index (EVI) and Net Primary Production (NPP).

- **Raw bands** from Landsat and Modis products.

- **Hydrography**: global water table, inundation and glaciers extent, and surface water change.

The long-term average and standard deviation of climatic variables and vegetation indices were computed from monthly data to capture their seasonal dynamics.

All covariates were projected to a common coordinate reference system (CRS), i.e. Goode's Homolosine projection for land
masses applied to the WGS84 *datum*. This projection was selected since among the equal-area projections supported by open source software it is the most effective minimising distortions over land (de Sousa et al., 2019). The projected covariates were imported to GRASS GIS in a normalised raster structure with cells of 250 m by 250 m. Covariates, and hence mapped areas, were restricted to land areas without built-up, water and glacier areas using a mask created from the ESA Land Cover layer for 2015 (Buchhorn et al., 2020). Thus properties of urban and subaqeous soils are not considered.

## 2.3 Covariates selection

Considering the large number of available environmental layers, a standardized and reproducible procedure to select covariates used for modelling was implemented to i) reduce redundancy between covariates, ii) obtain a more parsimonious and computationally-efficient model, iii) to decrease the risk of over-fitting (Gomes et al., 2019) and iv) to avoid a biased assessment of variable importance (Strobl et al., 2008). The covariates selection procedure consisted of two steps, de-correlation and
Recursive Feature Elimination.

### 2.3.1 De-correlation analysis

De-correlation analysis was carried out as initial step to reduce the redundancy of information from more than 400 environmental layers. Only covariate layers that had a pairwise correlation coefficient $<= 0.85$ with all other covariates were included in the subsequent analyses. For each pair of covariates correlated above this threshold, only the first one in alphabetical order
was selected for inclusion in the modelling phase. This step reduced the number of initial covariates to approximately 150 layers.

### 2.3.2 Recursive Feature Elimination

Recursive feature elimination (RFE) (Guyon et al., 2002) is a methodology that has proven effective to select an optimal set of covariates for regression trees models (Gomes et al., 2019; Hounkpatin et al., 2018). In this study, the RFE procedure
implemented in the `caret` package for the R language (Kuhn, 2015) was used, as it offers a good compromise between accuracy and computation time. The algorithm starts by fitting a model using all covariates, assessing its performance and ranking covariate importance. The least important covariates are then removed from the pool, and again the model is fitted,





assessed, and the least important covariates removed. The procedure repeats down to a pool between $0$ and $N$ covariates. This procedure is based on Out-Of-the-Bag (OOB) cross-validation and does not test all covariates combinations, but it is considered

one of the most robust covariates selection approaches for models like Random Forests (Nussbaum et al., 2018).

The RFE procedure on the full set of observations and covariates would prove computationally prohibitive. To improve computational feasibility for large datasets, additional steps were developed. Four sets of observations were used for RFE, each obtained using three cross-validation folds (see Section 2.1.3 for further details): set1 contained folds 1 to 3, set2 folds 4 to 6, set3 folds 7 to 9 and set4 contained fold 10 and 2 other random selected folds. In a first step, the RFE procedure from

`caret` was run independently on each set with default model hyper-parameters for `ranger` (i.e. *ntree* as 500 and *mtry* as the rounded square root of the number variables). In each set the optimal number and combination of covariates was automatically selected when the model performances stopped increasing, i.e. when the loss function reached its minimum. In this study, the loss function was the OOB RMSE.

In the second step, the RFE procedure was applied with all observations and all covariates selected in at least one of the four

sets used in the previous step. The final covariate set was the set minimising the loss function.

## 2.4 Hyper-parameter selection and cross-validation

Figure 2 summarises the approach used for the selection of the model hyper-parameters and the cross-validation. Further details are provided in the following sections.

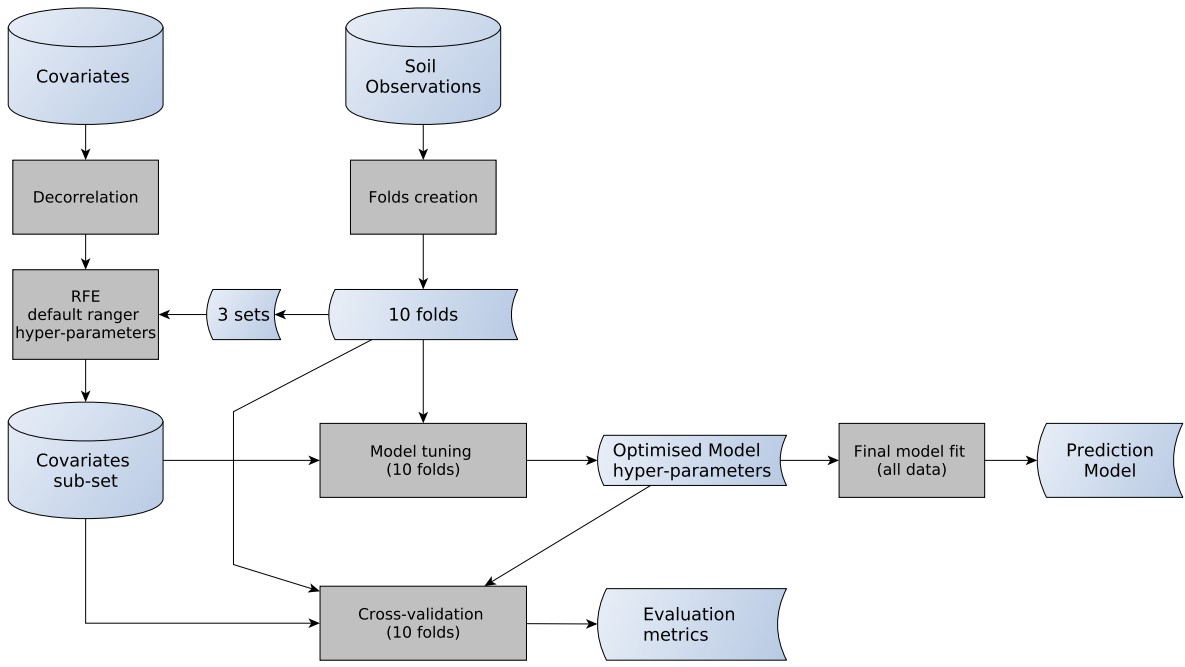

**Figure 2.** Detailed workflow for the Hyper-parameter selection and cross-validation.





### 2.4.1  Model tuning and validation

Model tuning was performed with a 10-fold cross-validation procedure applied to multiple combinations of hyper-parameters.

Different numbers of decision trees (*ntree* parameter) were combined with different numbers of covariates used in tree splits (*mtry* parameter). The number of trees were progressivly increased with the following values: 100, 150, 200, 250 500, 750 and 1000. The different *mtry* values were multiples of the square root of the number of covariates. Four multipliers were tested, 1 (default in `ranger`), 1.5, 2 and 3. For example, if the RFE procedure identified a set of 50 covariates, the *mtry* values assessed

were 7, 11, 14 and 21.

Each of the resulting combinations of *ntree* and *mtry* parameters was used to train a different model with observations from nine folds. Predictions were then assessed on the remaining fold with classical performance measures, i.e., root mean squared error (RMSE) and model efficiency coefficient (MEC;  Janssen and Heuberger, 1995). MEC is equal to the fraction of the explained variance based on the 1:1 line of predicted versus observed that is defined as 1 minus the ratio between residual sum

of squares and total sum of squares. The final hyper-parameter selection was based on an optimisation of model performance and computational constraints, in this case memory consumption. For example an increase of the *ntree* parameter above 200 provided a minor increment in the metrics (usually less than 0.1%, not reported here) while requiring considerably more memory and computation time.

The model evaluation was based on the performance metrics of the selected hyper-parameters combination.

## 2.5   Prediction and Uncertainty quantification

### 2.5.1   Model fit

The final model for each soil property was fitted with all available observations, the covariates and the hyperparameters selected in the previous steps. Observation depth was included in the model as a covariate. It was calculated at the mid-point of the sampled layer or horizon.

Models were obtained with the `ranger` package (Wright and Ziegler, 2017), with the option `quantreg` to build Quantile Random Forests (QRF; Meinshausen, 2006). With this option the prediction is not a single value, e.g., the average of predictions from the group of decision trees in the random forest, but rather a cumulative probability distribution of the soil property at each location and depth.

For each property (see Table 1) and standard depth from the GlobalSoilMap specification (0-5, 5-15, 15-30, 30-60, 60-100

and 100-200 cm) four different values were computed to characterise this distribution: median (0.50 quantile, $q_{0.50}$), mean, 0.05 quantile ($q_{0.05}$), and 0.95 quantile ($q_{0.95}$), i.e. the lower and upper limits of a 90% prediction interval. This uncertainty interval is as described in the GlobalSoilMap specifications (Arrouays et al., 2014). The predictions were computed for the mid-point of the depth interval and considered constant for the whole depth interval.

In order to compute the prediction uncertainty for soil texture, the back-transformation was applied at the level of individual

tree predictions, and the quantiles of the tree prediction distributions obtained from the resulting values.





### 2.5.2 Uncertainty

The percentage of validation observations contained in the 0.9 prediction interval was calculated (Prediction Interval Coverage probability, PICP) (Shrestha and Solomatine, 2006). Ideally the PICP is close to 0.9, indicating that the uncertainty was correctly assessed. A PICP substantially greater than 0.9 suggests that the uncertainty was underestimated, a substantially smaller PICP indicates that it was overestimated.

Furthermore, to visualise the uncertainty as a map, the following indicators were calculated:

1. $90^{th}$ prediction interval (PI90)

$$PI90 = q_{0.95} - q_{0.05} \qquad (2)$$

2. ratio of the interquantile range over the median (Prediction Interval Ratio, PIR):

$$PIR = \frac{q_{0.95} - q_{0.05}}{q_{0.50}} \qquad (3)$$

### 2.6 Qualitative evaluation of spatial patterns

Expert judgement was used to evaluate the reasonableness of the maps, by comparing well-known spatial patterns at global, regional, and local scales with SoilGrids predictions (see subsection 3.4). Obviously these are not definitive evaluations, only indicative.

### 2.7 Software and computational framework

SoilGrids requires an intensive computational workflow, with numerous steps integrating different software. SoilGrids is entirely based on open source software, in particular: SLURM (Yoo et al., 2003) for job management, GRASS GIS (GRASS Development Team, 2020) for data and tiles management, and R statistical software (R Core Team, 2020) for model fitting and statistical analysis.

Predictions were computed in a high-performance computing cluster. A dynamic geographic tiling system was developed with GRASS GIS to maximise the use of memory for each job. Technical details on this parallelisation scheme are given in de Sousa et al. (2020).

The predictions were multiplied by a conversion factor of 10 or 100 to maintain the required precision while using integer type in the file geotiff to reduce space occupied on disk. Application of the conversion factor resulted in mapped layers with units differing from those of the input observations (see Table 1).

The total computation time with the selected covariates and hyper-parameters differed per property. On average, the complete computation of the 24 maps (mean and three quantiles for each of six standard depths) for a single property, including: (i) RFE, (ii) model training, and (iii) prediction, took approximately 1 500 CPU-hours. The prediction accounted for about two thirds of the total time.





**Table 2.** Number of observations per standard depth interval for each soil property. See Table 1 for abbreviations.

| Depth interval | BDOD | CEC | CFVO | N | pH | SOC | STF |
|---|---|---|---|---|---|---|---|
| 0 - 5 cm | 8122 | 20576 | 15541 | 27192 | 44049 | 48616 | 42983 |
| 5 - 15 cm | 19817 | 49463 | 66833 | 82856 | 146677 | 148918 | 155302 |
| 15 - 30 cm | 17819 | 40673 | 35254 | 39568 | 91326 | 91682 | 98659 |
| 30 - 60 cm | 27146 | 63444 | 56755 | 48804 | 141812 | 122338 | 140353 |
| 60 - 100 cm | 23130 | 58038 | 50912 | 36946 | 131172 | 102687 | 127073 |
| 100 - 200 cm | 23396 | 66236 | 49995 | 28135 | 129373 | 92327 | 116847 |

## 3 Results and discussion

### 3.1 Input soil observations

Table 2 breaks down the distribution of the legacy soil observations for each soil property by depth interval. Table B1, in Appendix B, shows the number of observations by bio-climatic region.

Figures 3 and 4 show examples of observation density of the soil calibration data for two soil properties, $pH_{water}$ and proportion of coarse fragments, that show a large difference in density.

As indicated, the number of observations for each property varies greatly with depth and bioclimatic region, with higher densities observed for North America and Europe (Batjes et al., 2020). Generally, there are more observations for agricultural areas.

This study considers standardised data for some 240 000 profiles, derived from WoSIS. This is over 60 000 more profiles than considered in the data compilation underpinning the preceding SoilGrids runs (Hengl et al., 2017b), thus providing substantial new information for calibration of the new global models. However, as indicated, there are still significant geographic gaps (e.g., arid regions, boreal regions, and 'forest' soils). Some of these are related to the physical remoteness or unaccessibility of some regions, while others are related to the fact that many soil datasets still are not or can not be shared for various reasons as described by Arrouays et al. (2017).

In the previous version of SoilGrids (Hengl et al., 2017b), synthetic observations were randomly placed in regions with few or no observations, e.g., the Sahara and the Arabian Peninsula. This approach is worth further exploring, including information derived from other regional datasets, expert opinion and by transfer learning from similar areas according to the *Homosoil* concept (Mallavan et al., 2010), which assumes similarity of soil-forming factors across regions. However, SoilGrids already implicitly incorporates the *Homosoil* concept, as long as there are sufficient observations in a given soil-forming environment anywhere in the world. Therefore, no synthetic observations ('pseudo-points') were included in this version of SoilGrids, also by a lack of confidence about the accuracy of the synthetic data.

In future studies, it will be relevant to identify beforehand areas of the world with a low observation density that are not yet represented by a high density of observations in other areas with similar soil-forming factors. A set of synthetic profiles could





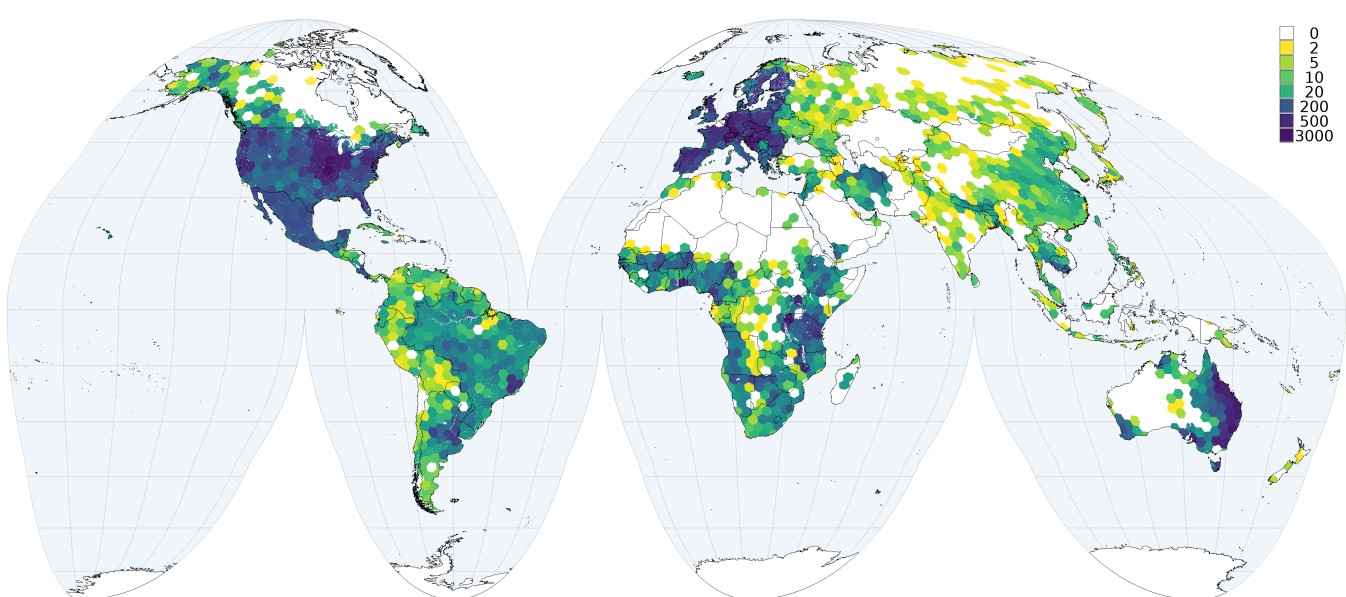

**Figure 3.** Number of observations per grid cell ( 70 000 $km^2$) for soil pH$_{water}$.

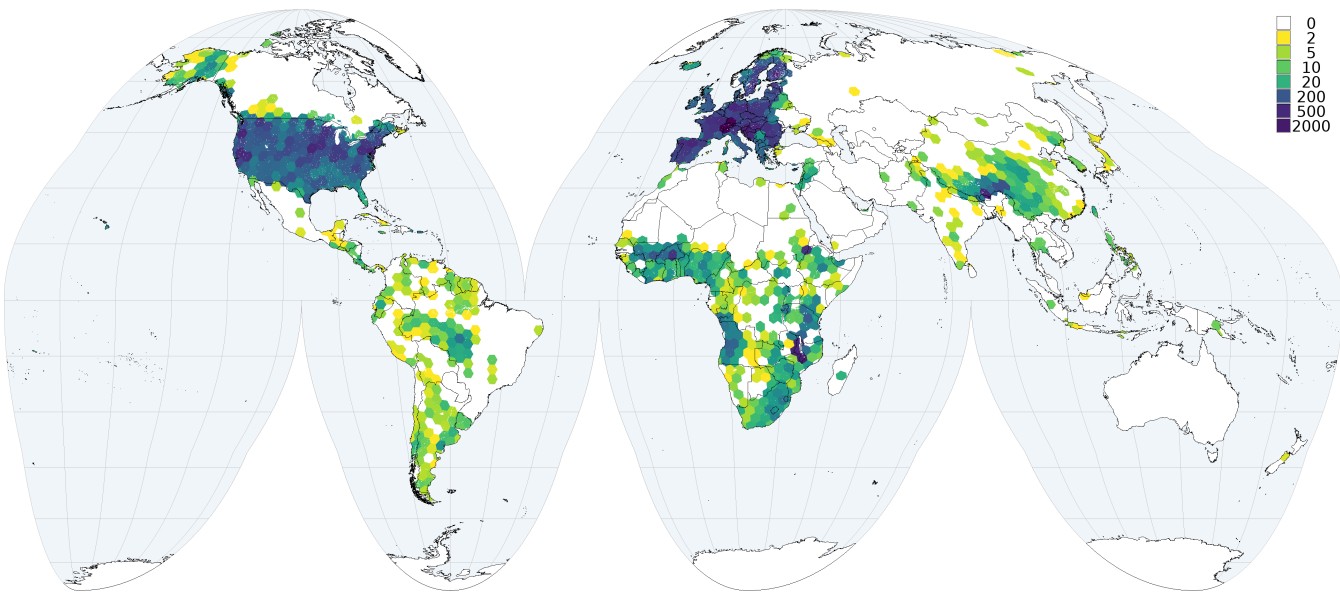

**Figure 4.** Number of of observations per grid cell ( 70 000 $km^2$) for coarse fragments.

then be generated to describe these areas, by consulting soil scientists knowledgeable on the soils and soil properties of these

areas.





## 3.2 Model tuning and hyper-parameters selection

Model hyper-parameters selected for each property are presented in Table 3.

The numbers of covariates selected using the two-step approach for covariates selection was fairly small in comparison with the full set (Table 3), resulting in more parsimonious models. Figure 5 shows two examples of the loss-function for RFE for
two soil properties with different number and distributions of input observations. In both cases there is a clear improvement of performances while using 15 to 20 covariates. The curve reaches a minimum of the loss function and then stays on a plateau with a slight decline after the identified minimum.

All final models were trained with a maximum of 200 decision trees, a number beyond which performance gains did not noticeably increase.

The *mtry* parameter mainly depended on the number of covariates and was always between 1.5 and 2 times the square root of the number of covariates, which is the default provided by common random forest packages such as *ranger* (Wright and Ziegler, 2017). This confirms the need to determine optimum model hyper-parameters, especially when dealing with large numbers of input data (Nussbaum et al., 2018) as is the case here.

**Table 3.** Hyper-parameters for each considered soil property. See Table 1 for abbreviations.

|  | Number of covariates | Number of trees | mtry |
|---|---|---|---|
| BDOD | 40 | 200 | 12 |
| CEC | 25 | 200 | 10 |
| CFVO | 20 | 200 | 6 |
| N | 30 | 200 | 10 |
| pH | 32 | 200 | 9 |
| SOC | 40 | 200 | 12 |
| Texture ALR I | 25 | 150 | 10 |
| Texture ALR II | 27 | 150 | 10 |

## 3.3 Quantitative evaluation

Cross-validation results are summarised in Table 4, presenting the root mean squared error (RMSE) and Model Efficiency Coefficient (MEC). The MEC varies from a minimum of 0.31 for coarse fragments to a maximum of 0.74 for BDOD. Clay is less well modelled than the other two particle-size classes. This may be an effect of the chosen ALR transformation that had clay as denominator (Lark and Bishop, 2007). Metrics of the mean were always better than or equal to those for the median for all properties.

Overall, these metrics are in line with continental or large regions DSM studies (Keskin and Grunwald, 2018). However, they are slightly lower than those preseneted by Hengl et al. (2017b). The latter difference can be explained by the more prudent



(a) pH

(b) CFVO

**Figure 5.** Loss function for RFE.





**Table 4.** Global cross-validation results for both mean and median predictions. See Table 1 for abbreviations.

| Property | RMSE (median) | RMSE (mean) | MEC (median) | MEC (mean) |
|---|---|---|---|---|
| BDOD | 0.19 | 0.19 | 0.73 | 0.74 |
| CEC | 11.01 | 10.69 | 0.40 | 0.43 |
| CFVO | 13.46 | 12.69 | 0.22 | 0.31 |
| N | 2.62 | 2.50 | 0.47 | 0.52 |
| pH | 0.78 | 0.77 | 0.67 | 0.68 |
| SOC | 39.67 | 36.48 | 0.37 | 0.47 |
| Sand | 0.19 | 0.18 | 0.51 | 0.54 |
| Silt | 0.13 | 0.13 | 0.60 | 0.62 |
| Clay | 0.13 | 0.13 | 0.42 | 0.43 |

**Table 5.** MEC per depth layer for mean predictions. See Table 1 for abbreviations.

| Depth Layer | BDOD | CEC | CFVO | N | pH | SOC | Sand | Silt | Clay |
|---|---|---|---|---|---|---|---|---|---|
| 0-5cm | 0.78 | 0.46 | 0.33 | 0.65 | 0.69 | 0.55 | 0.59 | 0.71 | 0.45 |
| 5-15cm | 0.74 | 0.42 | 0.35 | 0.41 | 0.66 | 0.39 | 0.58 | 0.64 | 0.42 |
| 15-30cm | 0.72 | 0.39 | 0.33 | 0.44 | 0.68 | 0.38 | 0.57 | 0.68 | 0.42 |
| 30-60cm | 0.70 | 0.42 | 0.31 | 0.46 | 0.68 | 0.38 | 0.54 | 0.62 | 0.41 |
| 60-100cm | 0.61 | 0.41 | 0.29 | 0.48 | 0.68 | 0.42 | 0.50 | 0.57 | 0.40 |
| 100-200cm | 0.59 | 0.45 | 0.29 | 0.49 | 0.67 | 0.59 | 0.48 | 0.54 | 0.40 |

cross-validation approach now taken, with spatially balanced folds and all observations belonging to the same profile in the same fold. This prevents using data from the same profile both for calibration and validation.

Table 4 shows that the models with a higher number of covariates have better predictive performances. However, these models are also the models with the largest number of observations (Table 2).

Table 5 shows the MEC for mean predictions by depth interval. Performances decreased with depth, in line with many other DSM studies (Keskin and Grunwald, 2018). This pattern can be explained in part by the reduced number of observations for deeper layers, but also by weakened relationships between environmental layers and soil properties of the deeper horizons.

In this study, the vertical dimension of soil variability was only taken into account by using the depth of the observation as a covariate. Alternatives such as 3D smoothers (Poggio and Gimona, 2017b) or geostatistical models exploiting 3D spatial auto-correlation are worth exploring in further studies.

Table 6 summarises the PICPs, globally and by predicted depth intervals. Most of the values are between 0.88 and 0.92, indicating that the predictions intervals obtained with QRF are a realistic representation of the prediction uncertainty, as the expected value for a 90% prediction interval is 0.90. Exceptions are the models for coarse fragments with higher values around




**Table 6.** Prediction Interval Coverage Probability, global and by predicted depth interval. See Table 1 for abbreviations.

| Property | Global | [0, 5] | [5, 15] | [15, 30] | [30, 60] | [60, 100] | [100, 200] |
|----------|--------|--------|---------|----------|----------|-----------|------------|
| BDOD | 0.90 | 0.89 | 0.91 | 0.91 | 0.91 | 0.90 | 0.88 |
| CEC | 0.88 | 0.89 | 0.90 | 0.88 | 0.88 | 0.88 | 0.87 |
| CFVO | 0.95 | 0.96 | 0.95 | 0.95 | 0.95 | 0.94 | 0.94 |
| N | 0.92 | 0.91 | 0.92 | 0.93 | 0.92 | 0.92 | 0.92 |
| pH | 0.90 | 0.91 | 0.91 | 0.90 | 0.91 | 0.90 | 0.89 |
| SOC | 0.92 | 0.91 | 0.92 | 0.92 | 0.92 | 0.92 | 0.92 |
| Sand | 0.79 | 0.82 | 0.82 | 0.80 | 0.78 | 0.78 | 0.78 |
| Silt | 0.96 | 0.95 | 0.96 | 0.96 | 0.96 | 0.96 | 0.96 |
| Clay | 0.96 | 0.96 | 0.96 | 0.96 | 0.95 | 0.95 | 0.96 |

0.95, indicating an overestimation of prediction uncertainty. The texture components have values with a larger spread, around 0.78 to 0.80 for sand and closer to 0.96 for silt and clay. These indicate a potential under-estimation of prediction intervals for sand and over-estimation for silt and clay. These results may be related with the range of these properties in the input observations. The transformation method used to derive the prediction intervals for the texture components could also be a contributing factor. Further exploration of the causes is worthwhile.

A key problem for DSM applications using legacy soil data is the evaluation ("validation") of the results. The best approach to numerical evaluation is to have an independent dataset obtained with probability sampling using a known sampling design (Brus et al., 2011; Brus, 2014). However, this is not feasible when only legacy data are available. In this case, a cross-validation approach is often used. Cross-validation needs to be tuned to avoid over- or under-estimation of the evaluation metrics, especially in case of large differences in observation density, i.e. clustered spatial observations. This is especially important at global

scale, as the distribution of the soil observations is not uniform across the globe. It can not be guaranteed that the evaluation metrics derived from cross-validation are unbiased estimates of the *true* validation metrics, i.e., those that would have been computed on a probability sample of the whole population. It is also not possible to quantify how close the cross-validation metrics estimates are to the true evaluation metrics, as it is not possible to obtain confidence intervals (Brus et al., 2011). When using cross-validation it is important to prevent over- or under-optimistic estimates. For example, it is likely that prediction

errors are smaller in areas where the sampling density is higher. Because of their high sampling density, such areas will be over-represented in the sample as the percentage of cross-validation points in clustered areas will be higher than the percentage of the total land area covered by those areas. Results of standard cross-validation will be strongly influenced by the performances in clustered areas. Using spatial cross-validation as suggested by Meyer et al. (e.g. 2018), where it is ensured that calibration data are never too close to a validation point, on the other hand could produce over-pessimistic results. In order to address

some of these concerns, this study followed a practical solution where the folds were created to guarantee a spatially balanced





distribution between validation folds, i.e., maintain the same densities of the input data in each fold so that they represent approximately the same population.

Although the numerical evaluation procedure used in this work takes into account the spatial distribution of the observations and their density, further improvement is necessary. For example, the weight assigned to heavily sampled areas could be re-335 duced. The USA and large regions of Europe and Australia have very high numbers of observations that could be reduced to further strengthen the spatial robustness of the validation procedure. Declustering or debiasing techniques (Goovaerts, 1997; Deutsch and Journel, 1998) have been applied with success in other geo-statistics exercises and could be adapted to the particular case of global soil mapping. The creation of the folds could also be modified to take into account the density of the observations.

### 340  3.4   Qualitative evaluation of spatial patterns

At global scale well-known patterns are reproduced, and typical properties associated with many World Reference Base for Soil Resources (WRB) (IUSS Working Group WRB, 2015) Reference Soil Groups can be recognised.

For example, the pH map identifies the large regions of alkaline soils (Solonetz, Solochak), highly-weathered soils (e.g., Acrisols, Alisols, Plithosols), acid forest soils (e.g., Podzols), and young soils from calcareous glacial deposits (e.g., Luvi-345 sols). The low pH of Andosols (e.g., Pacific Northwest USA, Japan, New Zealand) is also correctly represented. The texture components (PSC) maps correctly identify the siltier deltas (e.g., Yellow/Yangtze, Ganges/Brahmaputra), broad river plains (e.g., Po, Danube, Mississippi, Rio Plate, upper Amazon), the loess regions (e.g., midwestern USA, NW Europe, Ukraine), and the sandy North German/Polish plain. The cation exchange capacity (CEC) map clearly identifies large regions of highly-weathered clays (e.g., southeastern USA and China, central Brasil) and high-CEC 2:1 clays (e.g., "black cotton" Vertisols in 350 the Deccan plateau and the Sudan). This map, together with the soil organic carbon (SOC) concentration maps, identify large regions of Histosols (e.g., northern Canada, Scotland, Siberia, Borneo). The SOC stock map identifies deep Histosols and cool, wet regions (e.g., Pacific Northwest North American coast, Ireland, southern Chile). The coarse fragment map identifies large areas of the Tibetan plateau and the principal mountain chains, as well as recently-glaciated soils on igneous bedrocks (e.g., Scandinavia, northern Quebec and Ontario).

Many regional patterns are also clear, for example the pH transition from Sahara through Sahel to the West Africa coast, and the PSC transitions from the Des Moines glacial lobe to the pro-glacial loess deposits in Iowa (USA) as well as the PSC transition from clayey marine sediments along the North/Baltic sea coasts through the sandy plains to the central German loess belt. The CEC map identifies contrasting areas of Vertisols (e.g, "black belts" in Alabama/Mississippi and Texas USA). The coarse fragments map shows the detailed pattern of the basin-and-range region of the western USA and the ridge-and-valley 360 region of Appalachia.

However, at the local scale, a preliminary assessment of SoilGrids in the USA, comparing with a gridded version of the national detailed gSSURGO (NRCS National Soil Survey Center, 2016) soil geographic database based on detailed field survey, reveals that SoilGrids may fail to account for local parent material transitions, e.g., sedimentary facies of coastal plain





marine sediments, as well as glacial features such as proglacial lacustrine sediments and relic beach lines, so that the local PSC
pattern is not accurate, sometimes on the order of 20-30% of a particle-size class.

For example, Figure 6 (left) shows predicted sand concentration of the 0-5 cm layer in an approximately 50 x 50 km area in
central Pennsylvania (USA), ranging from approximately 25% (darker red) to 60% (darker blue). Important local differences
are clear: the low sand concentrations of the clayey soils in the limestone valleys trending SW-NE and the high concentrations
in soils from glacial till developed on sandstones in the north, as well as the residual soils on the resistant sandstone ridges of
the Ridge and Valley province of Appalachia in the south. These do generally agree with the detailed soil survey.

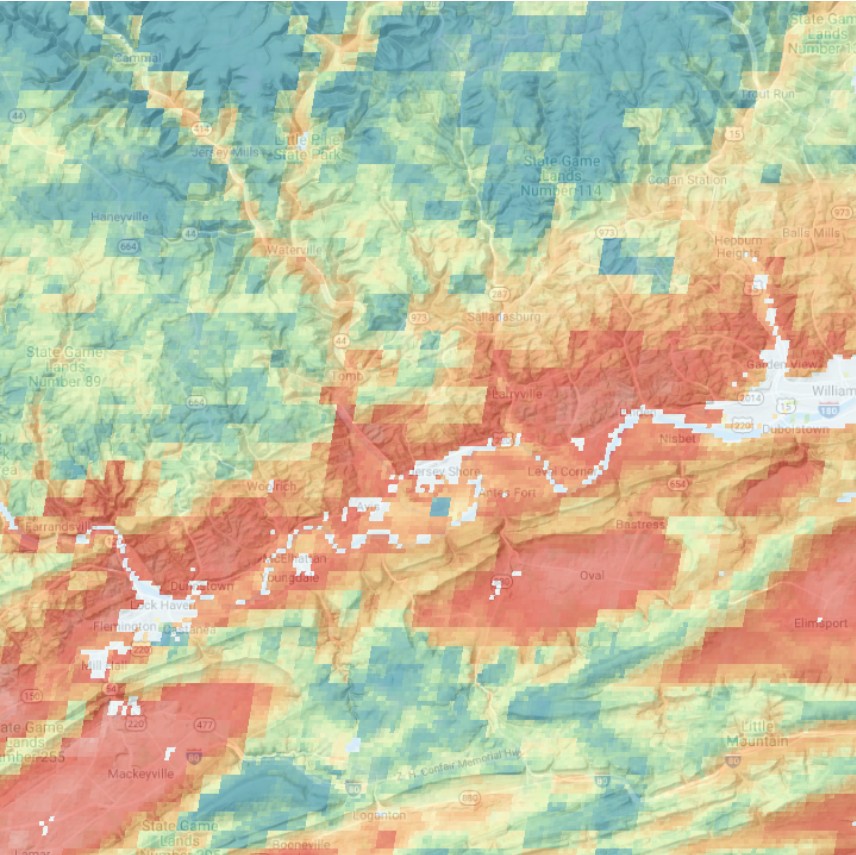

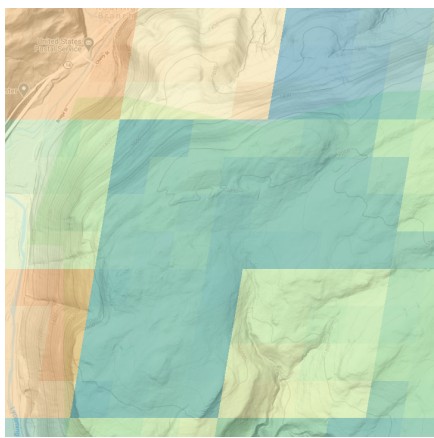

(a) Overview; centre $\approx -77° \ 14' \ E, 41° \ 14' \ N$, near Jersey Shore, PA

(b) Detail; centre $\approx -76° \ 56' \ E, 41° \ 33' \ N$,
southeast of Roaring Branch, PA

**Figure 6.** Predicted sand concentration, 0-5 cm, ground overlay in © Google Earth

At the detailed scale (250 m pixel) SoilGrids typically shows fine details that do not always appear to be related to obvious
landscape or land use differences, when the map is viewed as a ground overlay in Google Earth. For example, Figure 6 (right)
shows detail of predicted sand concentration of the 0-5 cm layer in an approximately 3 x 3 km area of the previous figure. The
effect of some covariates being at 1 km resolution and others at 250 m is apparent, but the reason for the fine-scale differences





is not. This area is of similar lithology, relief and land cover (second-growth dense forest) except the narrow valley at the
northwest edge, yet the predictions are quite different.

In this context, it should be realised that SoilGrids250m predictions are not meant for use at a detailed scale, i.e. at the
sub-national or local level, as national data providers often have access to more detailed point datasets and covariate layers for
their country than SoilGrids250m can consider (Chen et al., 2020; Roudier et al., 2020; Vitharana et al., 2019).

## 3.5  Prediction uncertainty

In general, the least sampled areas present the highest prediction uncertainties as expressed by the PICP. Figures 7 and 8 show
an example for two properties and depths (Maps for all properties and depths can be accessed at https://data.isric.org). Figure 9
shows an example representing the quantiles for $pH_{water}$ for the 60-100 cm layer. The north of Russia and the centre and north-
west of Canada are large regions for which few soil observations are available, therefore prediction distributions are wider than

in more densely sampled areas. However, these patterns are different for different properties. For example, arid areas actually
have the narrowest prediction ranges of $pH_{water}$. The uncertainty range is often wide for properties and regions with a wider
range of the property being modelled. This can be explained by the modelling approach performing more accurately within a
limited range of options. These regions also have larger local spatial variation with more difficulties for predictions.

The communication of uncertainty is an open challenge (Arrouays et al., 2020). Uncertainty should provide information for

policy makers and other stakeholders and not only scientists and modellers. The maps computed with eq. 3 are a first step in
this direction, but their limitations must be understood. For properties that have values at or near zero, e.g., coarse fragments,
they do not provide an entirely accurate uncertainty estimate. The use of uncertainty classes could be a further step to help
domain stakeholders.





(a) Prediction

(b) Inter-quantile range

**Figure 7.** Mean soil organic carbon content (dg/kg) prediction and range between 5% and 95% quantiles in the 5 cm to 15 cm depth interval.





(a) Prediction

(b) Uncertainty

**Figure 8.** Median total Nitrogen prediction (cg/kg) and associated uncertainty for the 15 cm to 30 cm depth interval.





(a) 5% quantile

(b) Mean

**Figure 9.** Prediction distribution for pH$_{water}$ (10*pH) in the 60 cm to 100 cm depth interval.





(c) Median

(d) 95% quantile

**Figure 9.** Prediction distribution for pH$_{water}$ (10*pH) in the 60 cm to 100 cm depth interval (cont.).



## 4   Conclusions and future work

The aim of this work was the production of quality-assessed global maps of soil properties, with cross-validation, hyper-parameters selection and quantification of uncertainty as implemented in the SoilGrids 2.0 product.

There are constant improvements in the DSM process, including the development and implementation of new methods and the use of new covariates that can help explain and model the spatial variation of soil properties. Products derived from Earth Observation are particularly relevant in this regard and have considerably improved over the last decade. For example,

the European Space Agency Sentinel missions (both optical and radar) provide high-resolution data that have been shown to improve DSM models performances.

Additional soil data for so far under-represented regions, for example the northern boreal regions as being collated by the International Soil Carbon Network (Malhotra et al., 2019), will be sought for possible consideration in the WoSIS workflow that provides the point data underpinning the SoilGrids mapping effort.

The use of decision tree-based models in DSM has become fairly common in recent years. Models such as Random Forests, XGBoost or Cubist tend to provide better results than most multiple linear regression methods with reasonable computation costs (Khaledian and Miller, 2020). However, methods such as Artificial Neural Networks promise further improvements in model performances if the amount and distribution of the data support these highly-complex models. This is the case in particular with convolutional or recursive neural networks (*deep learning*). However, these methods present computational

challenges with the amount of training data necessary for a sufficiently accurate DSM exercise, especially when working at global scale at medium to fine resolutions.

Cross-validation is another important aspect when considering how to assess and improve model performances. In particular, spatial cross-validation and declustering of the data need to be further explored.

This work described only the modelling of some of the primary soil properties, as defined and described in the Glob-

alSoilMap specifications. More work is necessary to obtain maps for soil thickness (rooting zone, solum or regolith), soil properties derived with pedo-transfer functions e.g. hydrological soil properties as saturated hydraulic conductivity (Pachepsky and Rawls, 2004) and complex properties that depend on multiple primary properties, e.g., carbon stocks. These layers are important inputs to model and map soil functions in the present and in the future as well as to support Earth System Modelling (Luo et al., 2016; Dai et al., 2019).

The quantification of uncertainty is recommended and is becoming more common in DSM studies; this work introduced it at global scale for the first time to our knowledge. While the provision of quantiles is mentioned in the GlobalSoilMap specifications, the representation and communication of uncertainty to end users and stakeholders remain an important research field to be further explored.

Finally, the integration of highly automatised workflows with expert opinion should be further explored. DSM products use

statistical models to describe soils and it is important to take into account the expertise and experience of pedologists, at least in an evaluation loop if not as part of the modelling itself.





*Code and data availability.* The code underpinning the SoilGrids 2.0 workflow is available under the GPL3 license at the SoilGrids git repository.

SoilGrids predictions themselves are available to the public under the Creative Commons CC-BY 4.0 licence, facilitating their widespread
use. They may be obtained as world mosaics in the Virtual Raster Tile (VRT) format from a WebDAV service. A suite of Web Coverage Services (WCS) facilitates automated access, e.g. from computer programmes or modelling frameworks. A set of notebooks was developed with examples for the use of the WCS. A new web based portal was also developed with this release, providing users with a light and swift means to visualise and explore the new predictions, making the best of state-of-the-art technologies for the web. A ReST API in beta stage is also available https://rest.soilgrids.org/.





**Appendix A: Environmental covariates**

Over four hundred geographic layers were available as environmental covariates for this work. These are chosen for their presumed relation to the major soil forming factors.

Table A1: Covariates sets and sources

**Weather and Climate**

• Temperature and precipitation from the Climatologies at high resolution for the earth's land surface areas (CHELSEA) dataset (Karger et al., 2016).

• Snowfall from ESA's CCI Land Cover dataset (Bontemps et al., 2013).

• Cloud cover by EarthEnv (Wilson and Jetz, 2016).

• Temperature and water vapour from NASA's MODIS products (Wan et al., 2006).

• Precipitation, solar radiation, temperature, water vapour, wind speed plus various indexes from the WorldClim version 2 climate data series (Fick and Hijmans, 2017).

**Ecology and ecosystems**

• Bioclimatic zones in the Global Ecophysiography product by the USGS Geosciences and Environmental Change Science Center (GECSC) (Dinerstein et al., 2017).

**Geology**

• Average soil and sedimentary-deposit thickness by the Distributed Active Archive Centre (DAAC) (Pelletier et al., 2016).

• Rock types by the USGS Geosciences and Environmental Change Science Center (GECSC), based on the Global Lithological Map database v1.1 (Hartmann and Moosdorf, 2012).

**Land Use and Land Cover**

• 2010 land cover classes from ESA's land cover CCI (Bontemps et al., 2013).

• Bare ground and tree cover from the USGU Global Land Cover dataset (Hansen et al., 2013).

• 2010 land cover classes from the NGCC's GLobeLand30 product (Chen et al., 2015).

**Elevation and morphology**

• Land surface elevation from the EarthEnv-DEM90 dataset (Robinson et al., 2014).

• Land surface elevation and various morphology indexes from the WorldGrids dataset (Reuter and Hengl, 2016).

• Land form classes in the Global Ecophysiography product by the USGS Geosciences and Environmental Change Science Center (GECSC) (Sayre et al., 2014).

**Core satellites outputs**





- Bands 3, 4, 5 and 7 from Landsat (Zanter, 2019)

- Middle and near infra-red bands from MODIS (Savtchenko et al., 2004).

**Vegetation Indexes**

- NDVI from Landsat (Zanter, 2019).

- EVI and NPP from MODIS (Savtchenko et al., 2004).

**Hydrography**

- Global Inundation Extent from Multi-Satellites (GIEMS) dataset by Estellus (Fluet-Chouinard et al., 2015).

- Extent of glaciers, surface water change and occurrence probability by the JRC (Pekel et al., 2016).

- Global water table depth (Fan et al., 2013).



## Appendix B: Bio-climatic regions

Table B1 summarises the number of observations per property for each bio-climatic region. An interactive map of the regions
is available on-line.

**Table B1.** Number of observations per property for each bio-climatic region. See Table 1 for abbreviations.

| Biome | CEC | CFVO | N | pH | SOC | STF |
|---|---|---|---|---|---|---|
| Tropical & Subtropical Moist Broadleaf Forests | 4185 | 2117 | 8378 | 12872 | 11901 | 11651 |
| Tropical & Subtropical Dry Broadleaf Forests | 558 | 205 | 1370 | 3264 | 2724 | 3051 |
| Tropical & Subtropical Coniferous Forests | 59 | 30 | 54 | 1336 | 878 | 1331 |
| Temperate Broadleaf & Mixed Forests | 12585 | 29708 | 24711 | 56569 | 49727 | 61822 |
| Temperate Conifer Forests | 6058 | 6417 | 5812 | 7597 | 9490 | 9834 |
| Boreal Forests/Taiga | 1443 | 3210 | 4834 | 4140 | 6819 | 5358 |
| Tropical & Subtropical Grasslands, Savannas & Shrublands | 8391 | 8259 | 20181 | 27633 | 24951 | 23135 |
| Temperate Grasslands, Savannas & Shrublands | 13442 | 9885 | 9812 | 23654 | 24421 | 25416 |
| Flooded Grasslands & Savannas | 246 | 124 | 503 | 754 | 818 | 798 |
| Montane Grasslands & Shrublands | 479 | 1865 | 1073 | 1386 | 3994 | 3568 |
| Tundra | 312 | 199 | 466 | 548 | 807 | 695 |
| Mediterranean Forests, Woodlands & Scrub | 1747 | 5951 | 8034 | 9126 | 12532 | 11428 |
| Deserts & Xeric Shrublands | 3342 | 3412 | 3224 | 8994 | 8163 | 9862 |
| Mangroves | 88 | 26 | 165 | 264 | 437 | 250 |

*Author contributions.* LP and LdS conceived and executed the research and wrote the paper. NB, GH, BK gave suggestions about the
approach and contributed extensively to the paper. DR designed and executed the qualitative evaluation and wrote the corresponding sections
in the paper. NB and ER designed and created the database of soil observations. All authors reviewed the paper.

*Competing interests.* The authors declare that they have no conflict of interest.

*Acknowledgements.* This work was funded from ISRIC core funding, with additional support from the EU-H2020 CIRCASA project. IS-
RIC – World Soil Information, legally registered as the International Soil Reference and Information Centre, receives core funding from the
Netherlands Government. We thank Rik van den Bosch (ISRIC Director) for the internal support to the project. We specially thank the or-
ganizations and experts that provided soil point data for consideration in WoSIS and SoilGrids. LP is member of a consortium supported by
LE STUDIUM Loire Valley Institute for Advanced Studies through its LE STUDIUM Research Consortium Programme.



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
