# Peer review of "SoilGrids 2.0: producing soil information for the globe with quantified spatial uncertainty"

_SOIL, 2020_

## Referee Comment (RC1) · Dominique Arrouays (Referee) · 10 Nov 2020

Review of the paper SoilGrids 2.0

This paper shows maps of soil properties for the entire globe at medium spatial resolution (250 metres cell size) using state-of-the-art machine learning methods to generate the necessary models. It takes as inputs soil observations from about 240 000 locations worldwide and over 400 global environmental covariates describing vegetation, terrain morphology, climate, geology and hydrology. The aim of this work was the production of quality-assessed global maps of soil properties, with cross-validation, hyper-parameters selection and quantification of spatially explicit uncertainty.

The main improvement compared to the previously published paper by Hengl et al., is

that the quality of the maps is assessed with cross-validation and quantification of spatially explicit uncertainty. This steps were missing in the previous versions of SoilGrids. Therefore, this is a huge progress that merits publication in SOIL.

I have only a few minor comments on the MS.

L43-44. "DSM consists primarily in building a qualitative numerical model between soil observations and environmental information acting as proxies for the soil forming factors." Not only, DSM may also use information acting directly as a proxy for a soil property (see for instance proximal sensing, remote sensing of bare soils, etc.)

L49 put e.g., "country (e.g. Mora-Vallejo….)" or enlarge the list of citations, many countries are omitted: e.g., Australia, US, Denmark, France, China, India, South Korea…)

L90. Not sure EU-Lucas (2013) can be considered as a soil profile DB. This is only topsoil, isn'it? May be find another way to say this.

L. 93-94. The readers would be interested in knowing briefly what were these minor corrections.

L. 131-132. Not sure this is clear for all the readers, a scheme or a flowchart would be useful.

L. 147-148. "The long-term average and standard deviation of climatic variables and vegetation indices were computed from monthly data to capture their seasonal dynamics." Be more precise: how long? From which date to which date?

L. 164. Why alphabetical order?

At the end of section 2.3.2 we would like to know how many covariates finally remained – this indicated in further tables but you could say that a number of covariates ranging between XX and xx were retained depending on the soil property.

Table 2. I'm very surprised to see that sometimes the number of observations increases with depth. Logically it should be the reverse, no?

Table 4. Missing units.

L. 299-300. So what? Needs a discussion. What is the most important? Think there is a paper by Samuel Rosa et al in Geoderma discussing these effects of the nb of covariates and the nb of points.

L. 304-305. This seems contradictory with the observation on table 2 (see before), I believe that "weakened relationships between environmental layers and soil properties of the deeper horizons" is more likely.

L. 334 and further. "The USA and large regions of Europe and Australia have very high numbers of observations that could be reduced to further strengthen the spatial robustness of the validation procedure". That's true for validation, but you reduce the number of calibration points. Is not here a kind of trade-off between que quality of predictions and the quality/robustness of the evaluation of the performance of the validation?

Good discussions in section 3.4 and 3.5!

Lines 414-419. "This work described only the modelling of some of the primary soil properties, as defined and described in the GlobalSoilMap specifications. More work is necessary to obtain maps for soil thickness (rooting zone, solum or regolith), soil properties derived with pedo-transfer functions e.g. hydrological soil properties as saturated hydraulic conductivity (Pachepsky and Rawls, 2004) and complex properties that depend on multiple primary properties, e.g., carbon stocks. These layers are important inputs to model and map soil functions in the present and in the future as well as to support Earth System Modelling (Luo et al., 2016; Dai et al., 2019)". I think this is more discussion than conclusion, it should be seen as a limitation of the study, you should explain why these properties could not be predicted and suggest ways to improve the situation. Future progress on how to predict these parameters should be proposed or taken from the literature where they exist.

Overall a very nice piece of work that merits publication after minor changes and some

development in discussion. Looking forward to seeing it published in SOIL!

---

## Short Comment (SC1) · 16 Nov 2020

Undoubtedly a huge effort to pull together the data and execute the rather complex workflow of globally mapping a selected number of soil properties at 250m grid cell resolution. Authors are to be congratulated for pulling this difficult computational task off. The quality and clarity of writing needs no further improvement in my opinion.

To me this paper reads as a methods paper and in doing so, does not introduce any new approaches to my knowledge. This is not a negative comment as it is important these types of documents exist to explain how such soil mapping products are produced. In saying this though i think the paper comes across as rather mechanical and does not demonstrate any deep knowledge of the global distribution of soil phenomena, rather a deep insight into statistical models and the validation of these models. The rather short discussion on the qualitative assessment of the mapping seems like a token attempt to slot some soil science into the work in my opinion.

Much discussion is made of the promised improvements of DSM over time due to new modelling capabilities, data and covariates etc. However, no mention or analysis is made about the comparison with Version 1 SoilGrids. Is version 2 better or worse? where are the improvements if any etc. Probably some work to do here. Would like to see comparisons with other existing digital soil maps outside of the USA too for example in France, Australia, UK and Denmark as a few examples.

Some comments are made of the scale issues with SoilGrids and they are probably not reliable for detailed analysis at sub-national scales etc. In areas of data richness with already well-developed soil mapping whether it be digital or legacy, shouldn't much more thoughtful and integrative analysis be pursued to combine these better products into the global digital soil mapping? No doubt much investment has been made to develop these data rich soil mapping infrastructures, but the top down approach implemented in this study neglects to take these efforts into account in my opinion. The ultimate outcome of having a suite of candidate maps of the same soil attribute over a specified spatial extent to a map user is confusion. Many people think, why so many different maps of the same thing? If ISRIC feel they have the imprimatur to produce world soil maps than i think approaches for doing this should not only be more consultative and collaborative with the global soil mapping community but to recognize the efforts and investments already made in areas of data richness and integrate that knowledge into the global work. There is little doubt that these global products show their value in data poor landscapes. Perhaps ISRIC should concentrate on this issue rather than push aside the intensive efforts of organisations whom have invested heavily in their own soil mapping infrastructure. In any case, a desktop and relatively easy fix would be combinatorial approaches to combine existing mapping with the SoilGrids models. The engagement with other practitioners is much more difficult to pull off with case-inpoint being the GlobalSoilMap.net initiative, but any constructive attempt at this to me is much better than a myopic top down approach that appears to have favour with the authors of this paper.

---

## Short Comment (SC2) · 21 Nov 2020

Thank you to Dr. Poggio for her frank rebuttal to my short comment about the role of ISRIC in delivering consistent global soil information. ISRIC has had a long and important role in advancing the knowledge and understanding of soils around the world, and it is great this vision has not waned and that the latest technologies and approaches to mapping are being exploited to advance this understanding further.

My comments to the paper were more just an open question about whether the approach undertaken in SoilGrids Version 2 (and SoilGrids Version 1 for that matter) is the best approach to take for delivering information about the global distribution of soils.

The point that i wish to take issue within Dr. Poggio's response is "...But to give global

modelers a consistent and seamless product we have chosen to use consistent global models, rather than compile a patchwork of national products." From a cartographic perspective i understand the need for seamlessness and consistency in order to avoid the off-putting visual of a patchwork of mapping products stitched together. From a modelling perspective however, one is not concerned about visuals, but rather the efficacy and reliability of the data for which said models will ingest. Therefore my argument is that wouldn't it be best for the global assessment of soil resources that the best or most reliable information be used, and thus the role of ISRIC is to compile this (which they have done already) and figure out a way to deliver this seamlessly to the end-user or their target group which are the modellers? I take issue with the fact that in order to deliver a seamless global soil map that a point-based top down approach be used instead of integrating the much more information rich soils data repository which includes existing maps whose provenance could be legacy soil survey or digital soil mapping, particularly if these data provide a more accurate depiction of the soil variability. It makes little sense to me to offer a seemingly sub-standard product on the argument that it is seamless and consistent. Models that ingest this information is just going to give spurious outcomes and that is not good for reputation and is ultimately a lost opportunity in my opinion. Dr Poggio's comments to say that this (data fusion or integration) is on their (ISRIC's) radar is encouraging and acknowledgment that their current point-based top down approach is somewhat deficient. Any effort to fast-track work on these integrative approaches (which Dr Poggio describe one in particular) would be very much welcomed.

---

## Short Comment (SC3) · 17 Dec 2020

I commend De Sousa et al. (2020) for their progress on the SoilGrids soil mapping project. The authors must be lauded for communicating the performance of their machine learning-based mapping of soil variables in various depths. In my opinion, the authors should address a couple of concerns I have with their approach before the manuscript is published in SOIL.

1. De Sousa et al. (2020) have predicted SOC content, bulk density, and coarse fraction volume. These are the ingredients to predict SOC stocks (kg m-2) which is arguably the metric most people are interested in. Is there a reason why SOC stocks are not presented in this manuscript? Is the modelling efficiency too low to calculate

[Figure]

SOC stocks with good conscience (median modelling efficiencies of 0.37 for SOC and 0.22 for coarse fraction volume)? It should be communicated with the reader if the machine learning model is good enough to calculate SOC stocks.

2. In the ISRIC FAQs there is a section on SOC stocks:

https://www.isric.org/explore/soilgrids/faq-soilgrids#How_where_SOC_stock_maps_generated

In my opinion, this should not be in grey "literature" but be discussed in this paper. What are the SOC stocks globally in SoilGrids version 2017 and SoilGrids 2.0? Why do they differ? A discussion of both approaches would be helpful.

3. In the paper, it is stated that "'Litter layers' on top of minerals soils were excluded from further modelling". This leads to a severe underestimation of SOC stocks. Apart from changing conventions in soil science, why should 'litter layers' be excluded? Peat layers, however, are included? In my opinion, organic layers should be included in the training of the machine learning model. Alternatively, the authors could provide estimates on an extra litter layer.

4. In the FAQs, it is also written: The organic layers on top of mineral soils were removed from the calculations and models. The total global carbon stocks obtained with version 2 (599 Pg of carbon for 0 to 30cm) are more in line with other global estimates (see for example: Jackson et al, 2017, Table 2 and Scharlemann et al, 2014).

Is the inclusion of the organic layer the reason for very high SOC stocks in SoilGrids 2017?

5. The cross-validation procedure between SoilGrids 2.0 and SoilGrids 2017 has changed. If I understand it correctly, in SoilGrids 2017 10-fold-cross-validation on individual datapoints was used while in SoilGrids 2020 10-fold-cross-validation on individual profiles has been used. Is this correct? In my opinion, the difference between the previous and current approaches should be discussed.

6. Ploton et al. (2020) have recently shown the importance of proper spatial crossvalidation. In their case, they assessed the effect of spatial autocorrelation by two approaches. On the one hand, they constructed spatial folds on clustered observations; on the other hand, they used a spatial blocking approach. From the method section, it is unclear if a spatial cross-validation was performed. "Balanced spatial distribution within each validation fold" – does this mean that every fold has just the same spatial coverage but were geographically close profiles possibly mixed for training and cross-validation? I would recommend a schematic drawing on how the cross-validation folds were constructed to make it clear to the readers of SOIL, especially those who are not familiar with digital soil mapping.

7. In Figures 4 and 5, the spatial coverage of some training data is shown. It seems like for many variables global digital soil mapping is still "Predicting into unknown space" which is part of the title of a paper by Meyer and Pebesma (2020). This paper deals with mapping the area of applicability for machine-learning-based digital mapping projects. There is an R package associated with it https://github.com/HannaMeyer/AOA_CaseStudy. It would be great if an area of applicability maps could be provided for every soil variable presented.

8. Observation depth was included as a covariate in the machine learning model. How much of the explained variance is explained by soil depth for the different variables? Additionally, a discussion if the 3D mapping approach may lead to biased estimates due to the stepped nature of tree-based method would be helpful. Here is a relevant paper that should probably be discussed here: Predicting soil properties in 3D by Ma and Fajardo (2021)

References:

De Sousa LM, Poggio L, Batjes NH, Heuvelink GBM, Kempen B, Riberio E, Rossiter D (2020) SoilGrids 2.0: producing quality-assessed soil information for the globe. pp Page, Copernicus GmbH.

Ma Y, Fajardo M (2021) Predicting soil properties in 3D: Should depth be a covariate?

Geoderma, 383, 114794.

Meyer H, Pebesma E (2020) Predicting into unknown space? Estimating the area of applicability of spatial prediction models. arXiv preprint arXiv:2005.07939.

Ploton P, Mortier F, Réjou-Méchain M et al. (2020) Spatial validation reveals poor predictive performance of large-scale ecological mapping models. Nature Communications, 11.
* * *

---

## Referee Comment (RC2) · Feng Liu (Referee) · 14 Jan 2021

Mapping soil properties for the globe is a very big challenge currently. I am happy to see the significant new progress made on the soilgrids project. Overall, the manuscript is written and organized very well. Here are my comments for the authors' consideration.

About the title: I think that assessing the quality of the resulting soil information from digital soil mapping is a very common practice. The assessment can only be a cross valiation of prediction accuracy. The "quality-assessed" seems not an obvious difference from other works. I guess that the biggest progress with regards to the SoilGrids1.0 is the addition of uncertainty estimations. I suggest revising the tile as "...producing global soil informaiton with spatial uncertainty.

[Figure]

The use of the cross validation procedure based on spatial stratification can guarantee a balanced spatial distribution within each validation fold. It is an improvement. However, in the model calibration, the same imbalance problem of spatial distribution of soil observations was totally neglected, which may lead to biased predictive models for the mapping and consequently poor performance in areas with very limited samples or even without samples.

The soil observations used in the work were from legacy soil surveys conducted at least three decades ago I think. But, some data of environmental covarites were derived from remote sensing observations of recent years, for example, the land use/cover data. The inconsistence in time may have significant influence on the predictions of easy-to-change soil properties such as SOC, N and PH.

I understand that the general way of modelbuilding in this work is still the same as SoilGrids1.0 version of Hengl et al (2017), simple 3D approach, which just takes observation (mid-point of horizon) depth as a covariate. I know this way is convenient in operation, one time modelbuilding can produce a soil property map of any depth. But there are some issues. One is that samples with high correlation at a same (profile) location may violate statistical modelling principles and lead to bias. Another is that taking depth as covariate may complicate model and make the model failed to focus its resources on capturing details of soil spatial variation but mainly the trend. This would lower the quality of the soil prediction overall. Some case studies also found that this way may tend to produce unrealistic soil predictions (Ma et al, 2021 and Nauman and Duniway, 2019).

I suggest that the authors also used R2 to express the performance from cross validation because it is very commonly used in soil mapping community and would be convenient to compare with other soil prediction studies.

The section of "Conclusions and future work" should be rewritten. I have difficulty to find a conclusion of this work. This section is not brief and clear. I feel that the discussion

and points of this section is too general and not going towards the SoilGrids Project. Readers may get nothing from it. The first paragraph of this section (Lines395-396) repeated the aim that already stated in the introduction section and should be deleted. The second paragraph does not belong to this section and should be removed. It would be better to condense the fourth paragraph (Lines405-411) into say two sentences. The last paragraph (Lines424-426) stated a very general point which can be applicable to any digital soil mapping work. With or without it does not make sense. I suggest putting words on the new progress of SoilGrids2.0, limitations and what next version would look like.

I suggest adding a map to show the time of the legacy soil survey projects of different countries if the time is very different among different regions. It is importance information for readers and data users to know what time/period of soil status the soil maps actually reflect.

Some expressions are confusing. The word "Quality-assessed" was frequently and somewhat widely used in the manuscript, which may have different specific meanings. For example, "producing quality-assessed soil informtion", "quality-assessed soil profile data", "Following data quality assessment and control", "Ultimately, upon final consistency checks, the quality-assessed and standardised data". This would make readers confused. Another similar problem is the use of "standardised", for example, "standardised soil profile data", does the "standardised" mean that all profile data were converted to the GSM depth intervals, as we saw "...six standard depths intervals" and "... standard depth interval for each soil property" in the title of Table2.

Line 14: "up to date information on world soil resources...is required to address ...". SoilGrids was based on legacy soil samples and the produced soil maps in fact reflect the status of soil conditions at least three decades ago, which is not up to date soil information. This work cannot respond to the demand.

Lines 143-145: The time information of the remote sensing data is missing.

Lines 168-185: it is not clear that which model was used in the RFE processing.

Lines 217-218: "…considered constant for the whole depth interval". this practice is different from that of SoilGrids1.0 (Hengl et al 2017) which generated prediction of a depth interval through calculating mathematical integral over depths. Is it better?

Table 2: I do not understand this table, actually confused. I guess that the the depth intervals of the soil horizons data used in this work are not uniform among the profiles and mostly not same as the GSM depth intervals. So, how to assign a sample of for example 20-50cm to a GSM depth interval, 15-30 or 30-60? Or maybe the authors standarised all horizons data into GSM depth intervals before modelling, then get the number of observations of each depth interval.

Figure 5: one graph example is enough.

Table 5: I also do not understand this table. As mentioned above, the horizon observations are not uniform in depth intervals. How did you compared with the predicted values at 2.5cm (0-5cm) to calculate the performance metrics MEC?

Line259: What means the "standarised data"?

Lines299-300: what do you want to express? large observations and covariates lead to better predictive performance?

Line315: evaluation=validation?

Figure6: please add a legend to the maps

Line379: some representative papers may be useful for reference to illustrate your point about national soil mapping: Liu et al. 2020, High-resolution and three-dimensional mapping of soil texture of China, Geoderma, 2020, 361: 114061; Liu et al 2020, A soil colour map of China. Geoderma, 2020, 379: 114556

---

## Author Response (AR1)

**SoilGrids 2.0: producing quality-assessed soil information for the globe**
**Response to reviewers**

Luis M. de Sousa[1], Laura Poggio[1], Niels H. Batjes[1], Gerard B. M. Heuvelink[1], Bas Kempen[1], Eloi Ribeiro[1], and David Rossiter[1]

[1]ISRIC - World Soil Information - Wageningen (NL)

**Correspondence:** Laura Poggio (laura.poggio@isric.org)

**1   RC1 - Dominique Arrouays**

Thank you very much for your useful comments. Below you will find detailed answers.

> *L43-44. "DSM consists primarily in building a qualitative numerical model between soil observations and environmental information acting as proxies for the soil forming factors." Not only, DSM may also use information acting directly as a proxy for a soil property (see for instance proximal sensing, remote sensing of bare soils, etc.)*

The sentence has been modified to:

*DSM consists primarily in building a quantitative numerical model between soil observations and environmental information acting as proxies for the soil forming factors. DSM can also integrate direct information as proxies for soil properties, for example proximal sensing measurements.*

> *L49 put e.g., "country (e.g. Mora-Vallejo. . ..)" or enlarge the list of citations, many countries are omitted: e.g., Australia, US, Denmark, France, China, India, South Korea. . .)*

The manuscript has been modified to recognise that these are only examples. "e.g." has been added to all relevant references.

> *L90. Not sure EU-Lucas (2013) can be considered as a soil profile DB. This is only topsoil, isn't it? May be find another way to say this.*

The manuscript has been modified using soil observations database instead of soil profiles database.

> *L. 93-94. The readers would be interested in knowing briefly what were these minor corrections.*

The manuscript has been modified by adding some examples of the minor checks, "for example further depth congruence checks".

> *L. 131-132. Not sure this is clear for all the readers, a scheme or a flowchart would be useful.*

The text was expanded to make the procedure more clear. We do not think that a full flowchart is necessary to explain a 10-fold split of the observed data.

*L. 147-148. "The long-term average and standard deviation of climatic variables and vegetation indices were computed from monthly data to capture their seasonal dynamics." Be more precise: how long? From which date to which date?*

The manuscript has been modified to include this information:

*The average and standard deviation of climatic variables and vegetation indices over 15 years (2001 - 2015) were computed from monthly data to capture their seasonal dynamics.*

*L. 164. Why alphabetical order?*

We think it is not that important which of the two covariates is retained. We used the alphabetical order to mimic a random choice of the covariates. Please note that we could not retain the covariate that has the strongest bivariate correlation with the dependent variable because we ran the de-correlation analysis as a preliminary step and not separately for each of the soil properties considered.

*At the end of section 2.3.2 we would like to know how many covariates finally remained – this indicated in further tables but you could say that a number of covariates ranging between XX and xx were retained depending on the soil property.*

The results mentioned are already in Section 3.3. Moving the results here would mix the results with the methods. We think it is better to keep them separate.

*Table 2. I'm very surprised to see that sometimes the number of observations increases with depth. Logically it should be the reverse, no?*

We carefully checked the numbers in the table and they are correct. Please note that the thickness of the six standard depth intervals is not constant and increases with depth. This could explain that the number of observations increases with depth.

*Table 4. Missing units.*

The units are summarised in Table 1. A reference to Table 1 has been added in all relevant table captions in the revised manuscript.

*L. 299-300. So what? Needs a discussion. What is the most important? Think there is a paper by Samuel Rosa et al in Geoderma discussing these effects of the nb of covariates and the nb of points.*

Here we point out an interesting result from the table. We did not do a study varying the number of points and covariates to come to a general conclusion on this. The paper referred to is, we think Samuel-Rosa, A., Heuvelink, G. B. M., Vasques, G. M. and Anjos, L. H. C.: Do more detailed environmental covariates deliver more accurate soil maps?, Geoderma, 243–244, 214–227, https://doi.org/10.1016/j.geoderma.2014.12.017, 2015. But this paper deals with the spatial detail of the covariates rather than the number of them, or the number of points. We have added a brief explanation in the text to clarify why we point this out.

*L. 304-305. This seems contradictory with the observation on table 2 (see before), I believe that "weakened relationships between environmental layers and soil properties of the deeper horizons" is more likely.*

Thank you for spotting this inconsistency. The text has been modified to make this more clear and focusing on weakened relationships between environmental layers and soil properties of the deeper layers.

55     *L. 334 and further. "The USA and large regions of Europe and Australia have very high numbers of observations that could be reduced to further strengthen the spatial robustness of the validation procedure". That's true for validation, but you reduce the number of calibration points. Is not here a kind of trade-off between the quality of predictions and the quality/robustness of the evaluation of the performance of the validation?*

This was poorly-phrased. We should not suggest reducing the number of calibration (training) points; rather, they could be
60   weighted according to the degree of clustering (sampling density). The folds used for evaluation would be similarly weighted. We did not investigate this yet, it will be considered in future work. Some text detailing this was added to the text.

    *Lines 414-419. "This work described only the modelling of some of the primary soil properties, as defined and described in the GlobalSoilMap specifications. More work is necessary to obtain maps for soil thickness (rooting zone, solum or regolith), soil properties derived with pedo-transfer functions e.g. hydrological soil properties as saturated hydraulic conductivity (Pachepsky*
65     *and Rawls, 2004) and complex properties that depend on multiple primary properties, e.g., carbon stocks. These layers are important inputs to model and map soil functions in the present and in the future as well as to support Earth System Modelling (Luo et al., 2016; Dai et al., 2019)". I think this is more discussion than conclusion, it should be seen as a limitation of the study, you should explain why these properties could not be predicted and suggest ways to improve the situation. Future progress on how to predict these parameters should be proposed or taken from the literature where they exist.*

70   The conclusion section has been re-written. We agree that most of the content of the Conclusions section of the submitted version was indeed more discussion than conclusion. In the revision we moved these parts to the Discussion section.

**2   RC2 - Feng Liu**

Thank you very much for your useful comments. Below you will find detailed answers.

    *About the title: I think that assessing the quality of the resulting soil information from digital soil mapping is a very common*
75     *practice. The assessment can only be a cross validation of prediction accuracy. The "quality-assessed" seems not an obvious difference from other works. I guess that the biggest progress with regards to the SoilGrids1.0 is the addition of uncertainty estimations. I suggest revising the tile as ". . .producing global soil information with spatial uncertainty.*

Thank you for the suggestion, indeed this is a key point. We will change the title to: "SoilGrids 2.0: producing soil information for the globe with quantified spatial uncertainty".

80     *The use of the cross validation procedure based on spatial stratification can guarantee a balanced spatial distribution within each validation fold. It is an improvement. However, in the model calibration, the same imbalance problem of spatial distribution of soil observations was totally neglected, which may lead to biased predictive models for the mapping and consequently poor performance in areas with very limited samples or even without samples.*

This is an interesting point. We agree that large differences in sampling density may also affect the calibration and that it
85   would be worthwhile to develop methods that take differences in sampling densities into account when calibrating a RF model.

However, this was beyond the scope of the current work. Please note that it is far from obvious how the calibration should be modified, because for calibration the distribution of the points in feature space is more important than their distribution in geographic space. Note also the link to a comment of Reviewer 1, who mentioned that reducing the number of observations in densely sampled areas would negatively influence calibration (see comment Reviewer 1 to L. 334 and further). We addressed this highly relevant issue in the Discussion of the revised manuscript.

> *The soil observations used in the work were from legacy soil surveys conducted at least three decades ago I think. But, some data of environmental covarites were derived from remote sensing observations of recent years, for example, the land use/cover data. The inconsistence in time may have significant influence on the predictions of easy-to-change soil properties such as SOC, N and PH.*

Thank you for this observation, which particularly concerns soil properties that are readily affected by changes in land use or management practices, and ideally age of the observations should be taken into account (see Batjes et al. (2020), p. 303). However, for dynamic soil properties such as pH and soil organic matter content, we considered that the spatial variation will be much greater than the temporal variation. Not explicitly considering time, in this study, should not affect the predictions greatly. For the future, we will look at space-time mapping at global scale, elaborating on the work of Heuvelink et al. (2020) for Argentina. We included these important considerations in the Discussion (Section 3.1).

> *I understand that the general way of modelbuilding in this work is still the same as SoilGrids1.0 version of Hengl et al (2017), simple 3D approach, which just takes observation (mid-point of horizon) depth as a covariate. I know this way is convenient in operation, one time modelbuilding can produce a soil property map of any depth. But there are some issues. One is that samples with high correlation at a same (profile) location may violate statistical modelling principles and lead to bias. Another is that taking depth as covariate may complicate model and make the model failed to focus its resources on capturing details of soil spatial variation but mainly the trend. This would lower the quality of the soil prediction overall. Some case studies also found that this way may tend to produce unrealistic soil predictions (Ma et al, 2021 and Nauman and Duniway, 2019).*

We are aware of the limitations of using depth as covariate as described in the studies cited by the reviewer. We think using depth as covariate is a practical approach in global modelling. We absolutely agree that further research is needed to assess the implications, especially when using global legacy datasets with varying densities. We have added text to discuss this point at some length.

> *I suggest that the authors also used R2 to express the performance from cross validation because it is very commonly used in soil mapping community and would be convenient to compare with other soil prediction studies.*

We understand the point made by the reviewer but we purposely refrained from reporting the $R^2$ because this metric has caused a lot of confusion in the digital soil mapping literature. The problem is that $R^2$ can refer both to the coefficient of determination of a linear regression between predicted and observed (i.e., the square of the Pearson correlation coefficient) as well as to the amount of variance explained by the model. The first evaluates how close predicted and observed are to a fitted straight line, while the second evaluates how close they are to the 1:1 line. It is the latter that we are after in statistical validation, and this is properly assessed using the (Nash-Sutcliffe) Model Efficiency Coefficient (MEC). We believe that the

digital soil mapping community is increasingly aware of these issues and that it will not take long until the use of R2 becomes obsolete and the community uses the MEC instead.

*The section of "Conclusions and future work" should be rewritten. I have difficulty to find a conclusion of this work. This section is not brief and clear. I feel that the discussion and points of this section is too general and not going towards the SoilGrids Project. Readers may get nothing from it. The first paragraph of this section (Lines395-396) repeated the aim that already stated in the introduction section and should be deleted. The second paragraph does not belong to this section and should be removed. It would be better to condense the fourth paragraph (Lines405-411) into say two sentences. The last paragraph (Lines424-426) stated a very general point which can be applicable to any digital soil mapping work. With or without it does not make sense. I suggest putting words on the new progress of SoilGrids2.0, limitations and what next version would look like.*

The conclusion section has been re-written. Most of the content of the discussion for the submitted version was indeed part of the discussion and it has been moved out of the conclusion section into a discussion of limitations and future work.

*I suggest adding a map to show the time of the legacy soil survey projects of different countries if the time is very different among different regions. It is importance information for readers and data users to know what time/period of soil status the soil maps actually reflect.*

Thank you for this suggestion. However, considering the large number of profiles we are working with, such a map would not be legible. Therefore, we have addressed this by adding statistics about the time periods the different point data were collected. Please note that we have also added text in the discussion section (3.1), where we advocate the development of a global model for space-time modelling as a follow up to the present research.

*Some expressions are confusing. The word "Quality-assessed" was frequently and somewhat widely used in the manuscript, which may have different specific meanings. For example, "producing quality-assessed soil informtion", "quality-assessed soil profile data", "Following data quality assessment and control", "Ultimately, upon final consistency checks, the quality-assessed and standardised data". This would make readers confused. Another similar problem is the use of "standardised", for example, "standardised soil profile data", does the "standardised" mean that all profile data were converted to the GSM depth intervals, as we saw ". . .six standard depths intervals" and ". . . standard depth interval for each soil property" in the title of Table2.*

We used 'quality-assessed' to indicate the various stages of checking the heterogeneous source point data, including consistency checks on lat-lon and depth of horizon/layer, flagging of duplicate profiles, providing measures for geographic and attribute accuracy, as well as time stamps; we also checked for possible erroneous entries (i.e. min, max checks). Details are provided elsewhere (Batjes et al., 2020; Ribeiro et al., 2018).

Standardisation here refers to making the soil analytical data comparable using 'operational definitions' that describe key elements of each method (see Ribeiro et al. (2018)); it also includes standardisation of the units of measurements and geo-referencing of the point locations.

As such, standardisation as used in this paper does not refer to adhering to the GSM depth specifications, as this 'standarisation' is part of the mapping process itself.

We removed this source of possible confusion from the manuscript, and added some clarification to Section 2.1.

*Line 14: "up to date information on world soil resources. . .is required to address . . .". SoilGrids was based on legacy soil samples and the produced soil maps in fact reflect the status of soil conditions at least three decades ago, which is not up to date soil information. This work cannot respond to the demand.*

We see your point and accordingly have rephrased this as follows: " ... based on the currently 'best-available' (shared) soil profile data."

 *Lines 143-145: The time information of the remote sensing data is missing.*

The manuscript has been modified to include this information:

*The average and standard deviation of climatic variables and vegetation indices over 15 years (2001 - 2015) were computed from monthly data to capture their seasonal dynamics.*

*Lines 168-185: it is not clear that which model was used in the RFE processing.*

 We have modified the manuscript to explicitly mention the method used in the RFE:

*In a first step, the RFE procedure from* `caret` *was run independently on each set with default model hyper-parameters for RandomForests algorithm as implemented in the package* `ranger` *(i.e. ntree as 500 and mtry as the rounded square root of the number variables)*

*Lines 217-218: ". . .considered constant for the whole depth interval". this practice is different from that of SoilGrids1.0 (Hengl*
 *et al 2017) which generated prediction of a depth interval through calculating mathematical integral over depths. Is it better?*

The reviewer is correct that in the previous version of SoilGrids (Hengl et al. 2017) predictions were made at the boundaries between GlobalSoilMap standard depth intervals. In this version of SoilGrids we went back to our original approach used in SoilGrids1km (Hengl et al. 2014), by predicting at the centre of the standard depth intervals and using these predictions as predictions of the interval average (i.e., by assuming that soil properties are constant within the interval). Because of this there
 was no need for numerical integration. We do not know which approach is better, but we do know that the approach that we now use again is much more commonly adopted in digital soil mapping. Both approaches have deficiencies because they do not handle the differences in vertical support of soil observations and predictions very well. A more solid approach is presented in Orton et al. (2016, Geoderma 262, 174-186), but this is much too demanding for high-resolution global soil mapping using machine learning.

 *Table 2: I do not understand this table, actually confused. I guess that the the depth intervals of the soil horizons data used in this work are not uniform among the profiles and mostly not same as the GSM depth intervals. So, how to assign a sample of for example 20-50cm to a GSM depth interval, 15-30 or 30-60? Or maybe the authors standarised all horizons data into GSM depth intervals before modelling, then get the number of observations of each depth interval.*

We carefully checked the numbers in the table and they are correct. Please note that the thickness of the six standard depth
 intervals is not constant and increases with depth. See also our reply to a comment by RC1.

*Figure 5: one graph example is enough.*

Figure 5 has been modified to show only one graph. The caption is now more informative.

*Table 5: I also do not understand this table. As mentioned above, the horizon observations are not uniform in depth intervals. How did you compared with the predicted values at 2.5cm (0-5cm) to calculate the performance metrics MEC?*

The text has been expanded to explain how the cross-validation took into account the differences in depth.

*Line259: What means the "standarised data"?*

As indicated in an earlier reply, "Standardisation here refers to making the soil analytical data comparable using 'operational definitions' that describe key elements of each method (see Ribeiro et al. (2018)); it also includes standardisation of the units of measurements and geo-referencing of the point locations."

*Lines299-300: what do you want to express? large observations and covariates lead to better predictive performance?*

This was also pointed out by RC1. Here we point out an interesting result from the table. We did not do a study varying the number of points and covariates to come to a general conclusion on this. We have added a brief explanation in the text to clarify why we point this out.

*Line315: evaluation=validation?*

We have added a reference to the argument of Oreskes (1998) and Rossiter (2017) on this, also explained our preferred usage. We link to the common use of "validation" in these contexts, which we refer to as "numeric evaluation". We have checked the entire text for consistent usage of these terms.

*Figure6: please add a legend to the maps*

Added.

*Line379: some representative papers may be useful for reference to illustrate your point about national soil mapping: Liu et al. 2020, High-resolution and three-dimensional mapping of soil texture of China, Geoderma, 2020, 361: 114061; Liu et al 2020, A soil colour map of China. Geoderma, 2020, 379: 114556*

Thank you for the suggestion, indeed this fits into the list of papers we already cited for this point. Since no paper showing a regional map of China was included in that list, we have added the suggested reference.

**3 SC3 - Bernhard Ahrens**

Thank you very much for your useful comments. Below you will find detailed answers.

*I commend De Sousa et al. (2020) for their progress on the SoilGrids soil mapping project. The authors must be lauded for communicating the performance of their machine learning-based mapping of soil variables in various depths. In my opinion, the authors should address a couple of concerns I have with their approach before the manuscript is published in SOIL.*

*1. De Sousa et al. (2020) have predicted SOC content, bulk density, and coarse fraction volume. These are the ingredients to predict SOC stocks (kg m-2) which is arguably the metric most people are interested in. Is there a reason why SOC stocks are not presented in this manuscript? Is the modelling efficiency too low to calculate SOC stocks with good conscience (median modelling efficiencies of 0.37 for SOC and 0.22 for coarse fraction volume)? It should be communicated with the reader if the machine learning model is good enough to calculate SOC stocks.*

We are in the process of preparing a manuscript that will discuss and compare global estimates of SOC stocks. That study will derive SOC stocks from the point observations themselves, not from the various data layers. As such this topic is beyond the scope of the present paper.

*2. In the ISRIC FAQs there is a section on SOC stocks: https://www.isric.org/explore/soilgrids/faq-soilgrids#How_where_ SOC_stock_maps_generated. In my opinion, this should not be in grey "literature" but be discussed in this paper. What are the SOC stocks globally in SoilGrids version 2017 and SoilGrids 2.0? Why do they differ? A discussion of both approaches would be helpful.*

We find this a somewhat surprising comment as you refer to the FAQ section on our website. As indicated earlier, we are preparing a separate manuscript on SOC stocks, discussing differences in methodological approaches. This paper would have been too large to include all the modelling steps and decisions necessary for the production of global soil carbon stocks. Therefore we decided to focus only on primary soil properties as defined by the GlobalSoilMap specifications.

*3. In the paper, it is stated that "'Litter layers' on top of minerals soils were excluded from further modelling". This leads to a severe underestimation of SOC stocks. Apart from changing conventions in soil science, why should 'litter layers' be excluded? Peat layers, however, are included? In my opinion, organic layers should be included in the training of the machine learning model. Alternatively, the authors could provide estimates on an extra litter layer.*

Thanks for the useful suggestion. This is an element we are considering in the SOC manuscript, currently under preparation.

*4. In the FAQs, it is also written: The organic layers on top of mineral soils were removed from the calculations and models. The total global carbon stocks obtained with version 2 (599 Pg of carbon for 0 to 30cm) are more in line with other global estimates (see for example: Jackson et al, 2017, Table 2 and Scharlemann et al, 2014). Is the inclusion of the organic layer the reason for very high SOC stocks in SoilGrids 2017?*

Again, you are referring to the FAQ pages on our website. The present paper is not a comparison of the SG2017 and SG2020 predictions. Possibly, we may consider this intriguing aspect in the upcoming SOC manuscript, mentioned above.

*5. The cross-validation procedure between SoilGrids 2.0 and SoilGrids 2017 has changed. If I understand it correctly, in SoilGrids 2017 10-fold-cross-validation on individual datapoints was used while in SoilGrids 2020 10-fold-cross-validation on individual profiles has been used. Is this correct? In my opinion, the difference between the previous and current approaches should be discussed.*

The reviewer is right that we modified the k-fold cross-validation procedure by ensuring that all observations belonging to the same profile were in the same fold. In addition, we did not randomly assign profiles to folds but applied a strategy that enforces that the folds are spatially balanced. We modified the text in the manuscript to include additional details:

*Profile locations were stratified into a quasi-regular partition of the Earth's surface. An Icosahedral Snyder Equal-Area Grid (ISEAG) of aperture 3 and resolution 6 was created for this purpose with the dggridR package for the R language (Barnes et al., 2016). This grid consists of 7 292 cells (mostly hexagons) of an average area close to 70 000 km2. Each observation was assigned to the ISEAG cell (stratum) in which the corresponding soil profile is located.*

*The observations assigned to a spatial stratum were distributed evenly among the ten validation folds using the caret package for the R language. In this way each spatial stratum contributed a similar number of observations to each of the validation folds. All observations were kept in the same fold for both model calibration and evaluation.*

*6. Ploton et al. (2020) have recently shown the importance of proper spatial cross validation. In their case, they assessed the effect of spatial autocorrelation by two approaches. On the one hand, they constructed spatial folds on clustered observations; on the other hand, they used a spatial blocking approach. From the method section, it is unclear if a spatial cross-validation was performed. "Balanced spatial distribution within each validation fold" – does this mean that every fold has just the same spatial coverage but were geographically close profiles possibly mixed for training and crossvalidation? I would recommend a schematic drawing on how the cross-validation folds were constructed to make it clear to the readers of SOIL, especially those who are not familiar with digital soil mapping*

As explained in our reply to your previous comment we explained the spatially balanced cross-validation approach in more detail in the revision (but we prefer to not include a schematic drawing). Our approach does not belong to the class of 'spatial cross-validation' techniques as used in Ploton et al. (2020). The reason that we did not use such technique is that we think these methods have no theoretical underpinning and produce biased (i.e., overpessimistic) validation results. In fact, one of the authors has a 'Matters Arising' paper under review that addresses the misconceptions of the Ploton et al. (2020) paper.

*7. In Figures 4 and 5, the spatial coverage of some training data is shown. It seems like for many variables global digital soil mapping is still "Predicting into unknown space" which is part of the title of a paper by Meyer and Pebesma (2020). That paper deals with mapping the area of applicability for machine learning-based digital mapping projects. There is an R package associated with it https://github.com/HannaMeyer/AOA_CaseStudy. It would be great if an area of applicability maps could be provided for every soil variable presented.*

Thank you very much for raising this important point. We are aware of the paper from Meyer and Pebesma (2020). We realise its implications for DSM, especially when spanning across large areas with varying densities of input observations. We are exploring this approach for future improvements of the procedure. We added a mention to these issues into the discussion of the paper.

*8. Observation depth was included as a covariate in the machine learning model. How much of the explained variance is explained by soil depth for the different variables? Additionally, a discussion if the 3D mapping approach may lead to biased estimates due to the stepped nature of tree-based method would be helpful. Here is a relevant paper that should probably be discussed here: Predicting soil properties in 3D by Ma et al. (2021).*

We are aware of the limitations of using depth as covariate (2.5 DSM) described in the recent studies mentioned. We think using depth as covariate is a practical approach in global modelling. We absolutely agree that further research is needed to assess the implications especially when using global legacy datasets with varying densities. See also our reply to a comment by RC2. We have modified the text to discuss this point:

*In this study, the vertical dimension of soil variability was only taken into account by using the depth of the observation as a covariate (so-called "2.5 DSM"). Recent publications (Ma et al., 2021; Nauman and Duniway, 2019) indicate that such approach can be too simplistic or lead to problems with consistency over the predicted depth sequence. This may be true for*

*local datasets where the short-range spatial variability is of a similar magnitude as the vertical variability. Further research is necessary to assess the effects of using depth as a covariate on global datasets and models. Alternatives such as 3D smoothers (Poggio and Gimona, 2017) or geostatistical models exploiting 3D spatial auto-correlation are worth exploring in further studies.*

*References: De Sousa LM, Poggio L, Batjes NH, Heuvelink GBM, Kempen B, Riberio E, Rossiter D (2020) SoilGrids 2.0: producing quality-assessed soil information for the globe. pp Page, Copernicus GmbH.*

*Ma Y, Fajardo M (2021) Predicting soil properties in 3D: Should depth be a covariate? Geoderma, 383, 114794.*

*Meyer H, Pebesma E (2020) Predicting into unknown space? Estimating the area of applicability of spatial prediction models. arXiv preprint arXiv:2005.07939.*

*Ploton P, Mortier F, Réjou-Méchain M et al. (2020) Spatial validation reveals poor predictive performance of large-scale ecological mapping models. Nature Communications, 11.*

**4   SC1 Brendan Malone**

*Undoubtedly a huge effort to pull together the data and execute the rather complex workflow of globally mapping a selected number of soil properties at 250m grid cell resolution. Authors are to be congratulated for pulling this difficult computational task off. The quality and clarity of writing needs no further improvement in my opinion. To me this paper reads as a methods paper and in doing so, does not introduce any new approaches to my knowledge. This is not a negative comment as it is important these types of documents exist to explain how such soil mapping products are produced. In saying this though i think the paper comes across as rather mechanical and does not demonstrate any deep knowledge of the global distribution of soil phenomena, rather a deep insight into statistical models and the validation of these models. The rather short discussion on the qualitative assessment of the mapping seems like a token attempt to slot some soil science into the work in my opinion. Much discussion is made of the promised improvements of DSM over time due to new modelling capabilities, data and covariates etc. However, no mention or analysis is made about the comparison with Version 1 SoilGrids. Is version 2 better or worse? where are the improvements if any etc. Probably some work to do here. Would like to see comparisons with other existing digital soil maps outside of the USA too for example in France, Australia, UK and Denmark as a few examples. Some comments are made of the scale issues with SoilGrids and they are probably not reliable for detailed analysis at sub-national scales etc. In areas of data richness with already well-developed soil mapping whether it be digital or legacy, shouldn't much more thoughtful and integrative analysis be pursued to combine these better products into the global digital soil mapping? No doubt much investment has been made to develop these data rich soil mapping infrastructures, but the top down approach implemented in this study neglects to take these efforts into account in my opinion. The ultimate outcome of having a suite of candidate maps of the same soil attribute over a specified spatial extent to a map user is confusion. Many people think, why so many different maps of the same thing? If ISRIC feel they have the imprimatur to produce world soil maps than i think approaches for doing this should not only be more consultative and collaborative with the global soil mapping community but to recognize the efforts and investments already made in areas of data richness and integrate that knowledge into the global work. There is little doubt that these global products show their value in data poor landscapes. Perhaps ISRIC should concentrate on this issue rather than push aside the intensive efforts of organisations whom have invested heavily in their own soil mapping infrastructure. In any case, a desktop and relatively easy fix would be combinatorial approaches to combine existing mapping*

*with the SoilGrids models. The engagement with other practitioners is much more difficult to pull off with case-in point being*

325 *the GlobalSoilMap.net initiative, but any constructive attempt at this to me is much better than a myopic top down approach*

*that appears to have favour with the authors of this paper.*

**Published response**

*We thank Dr. Malone for his insightful comments on our manuscript 'SoilGrids 2.0: producing quality-assessed soil informa-*

*tion for the globe'. Besides various suggestions for improving the product itself, which we will address later in our combined*

330 *comments to the two referee reports, we would like to respond here to the fourth paragraph of the interactive comment. This*

*paragraph directly addresses ISRIC's role and approach. Overall, we agree with the observation that SoilGrids250m predic-*

*tions are not to be recommended for 'detailed' use at sub-national scale. To that end we provide measures for uncertainty*

*in the predictions, as resolution itself does not convey any information on (regional) accuracy. In the manuscript, we indi-*

*cate this clearly by stating: "In this context, it should be realized that SoilGrids250m predictions are not meant for use at*

335 *a detailed scale, i.e. at the sub-national or local level, as national data providers often have access to more detailed point*

*datasets and covariate layers for their country than SoilGrids250m can consider (Chen et al., 2020; Roudier et al., 2020;*

*Vitharana et al., 2019). The advanced work in the USA and Australia may be added to the examples. Coming back to IS-*

*RIC's role. ISRIC in no sense is "push[ing] aside the intensive efforts of organisations [which] have invested heavily in*

*their own soil mapping infrastructure." On our website we compile all publically-available sources of digital soil information*

340 *(see https://www.isric.org/explore/soil-geographic-databases) and have consistently recommended that users in countries with*

*high-quality products use those products for their soil use and management needs. We are not only acknowledging the fact*

*that nationally produced products are preferred for national level applications, we also stimulate the production of national*

*level products through cooperation with national soil institutes. From its inception, following up on the recommendation of the*

*International Union of Soil Sciences (IUSS, then ISSS) and the UNESCO General Council, ISRIC was created to support the*

345 *FAO-UNESCO-ISSS "Soil-Map-of-the-World-Project" as started in 1961. Since that time, ISRIC has been involved with the*

*development of broadscale soil databases, initially using a pedologybased approach (e.g. SOTER, WISE and HWSD, as refer-*

*enced in the manuscript). By their nature, these projects involved international collaboration and capacity building. Presently,*

*as an example of worldwide collaboration, we are co-developing the federated Global Soil Information System of the Global*

*Soil Partnership (GSP) aimed at a bottom-up approach, with intensive country collaboration and capacity building foreseen.*

350 *Our global soil mapping effort are complementary to the above-mentioned bottomup programs. Complementary in the sense*

*that the products developed by the national partners are applied at national and sub national level by national stakeholders,*

*whereas SoilGrids is a globally-consistent and seamless product to be used by (e.g.) global environmental modellers. SoilGrids*

*is meant as a consistent product, not as a replacement for good national products. But to give global modellers a consistent and*

*seamless product we have chosen to use consistent global models, rather than compile a patchwork of national products. While*

355 *developing SoilGrids we do collaborate intensively with many different data providers and with global experts for the design of*

*the methodology, e.g. as active member of the GlobalSoilMap.net consortium, now a working group of the International Union*

*of Soil Sciences and GSP partner. The suggestion to use existing national and regional gridded products in order to improve*

*global products is a valuable suggestion and on our list of future developments. The difficulty is the patchwork nature of the*

*coverage. We plan to investigate how to best realise such integration, perhaps using these products as priors in a Bayesian*

360 *approach. This is an active area of research and outside the scope of the current productdescribed in this paper. We certainly*

*look forward to further collaboration with Dr Malone's and other DSM groups, enabling an environment for collaborative research.*

**5 SC2 - Brendan Malone**

*Thank you to Dr. Poggio for her frank rebuttal to my short comment about the role of ISRIC in delivering consistent global soil information. ISRIC has had a long and important role in advancing the knowledge and understanding of soils around the world, and it is great this vision has not waned and that the latest technologies and approaches to mapping are being exploited to advance this understanding further. My comments to the paper were more just an open question about whether the approach undertaken in SoilGrids Version 2 (and SoilGrids Version 1 for that matter) is the best approach to take for delivering information about the global distribution of soils. The point that i wish to take issue within Dr. Poggio's response is "...But to give global modelers a consistent and seamless product we have chosen to use consistent global models, rather than compile a patchwork of national products." From a cartographic perspective i understand the need for seamlessness and consistency in order to avoid the off-putting visual of a patchwork of mapping products stitched together. From a modelling perspective however, one is not concerned about visuals, but rather the efficacy and reliability of the data for which said models will ingest. Therefore my argument is that wouldn't it be best for the global assessment of soil resources that the best or most reliable information be used, and thus the role of ISRIC is to compile this (which they have done already) and figure out a way to deliver this seamlessly to the enduser or their target group which are the modellers? I take issue with the fact that in order to deliver a seamless global soil map that a point-based top down approach be used instead of integrating the much more information rich soils data repository which includes existing maps whose provenance could be legacy soil survey or digital soil mapping, particularly if these data provide a more accurate depiction of the soil variability. It makes little sense to me to offer a seemingly sub-standard product on the argument that it is seamless and consistent. Models that ingest this information is just going to give spurious outcomes and that is not good for reputation and is ultimately a lost opportunity in my opinion. Dr Poggio's comments to say that this (data fusion or integration) is on their (ISRIC's) radar is encouraging and acknowledgment that their current point-based top down approach is somewhat deficient. Any effort to fast-track work on these integrative approaches (which Dr Poggio describe one in particular) would be very much welcomed.*

We appreciate Dr. Malone's reply and clarification. Most of his points are covered in our original response. We here point out that ISRIC-World Soil Information in fact does support the "bottom-up" integration of nationally-developed digital soil maps via the GLOSIS (Global Soil Information Systems) initiative of the Global Soil Partnership; see https://www.glosis.org/ and https://glosis.isric.org/. So this is a parallel track, integrating information provided by national partners, to the SoilGrids track of a globally-consistent product, following the "Homosoil" theory as explained in the Introduction of our paper. We encourage users to integrate SoilGrids as prior information in their own DSM exercises, e.g., as a covariate layer to combine with their local covariates. We still feel that a product with a consistent basis may be preferred by global modellers for certain applications. But it is by no means an either-or question.

**References**

Batjes, N. H., Ribeiro, E., and van Oostrum, A.: Standardised soil profile data to support global mapping and modelling (WoSIS snapshot 2019), Earth System Science Data, 2020, 299–320, https://doi.org/10.5194/essd-12-299-2020, https://doi.org/10.5194/essd-12-299-2020, 2020.

Heuvelink, G. B. M., Angelini, M. E., Poggio, L., Bai, Z., Batjes, N. H., van den Bosch, R., Bossio, D., Estella, S., Lehmann, J., Olmedo, G. F., and Sanderman, J.: Machine learning in space and time for modelling soil organic carbon change, European Journal of Soil Science, n/a, https://doi.org/https://doi.org/10.1111/ejss.12998, 2020.

Ma, Y., Minasny, B., McBratney, A., Poggio, L., and Fajardo, M.: Predicting soil properties in 3D: Should depth be a covariate?, Geoderma, 383, 114 794, https://doi.org/https://doi.org/10.1016/j.geoderma.2020.114794, http://www.sciencedirect.com/science/article/pii/S0016706120325490, 2021.

Meyer, H. and Pebesma, E.: Predicting into Unknown Space? Estimating the Area of Applicability of Spatial Prediction Models, arXiv:2005.07939 [cs, stat], http://arxiv.org/abs/2005.07939, 2020.

Nauman, T. W. and Duniway, M. C.: Relative prediction intervals reveal larger uncertainty in 3D approaches to predictive digital soil mapping of soil properties with legacy data, Geoderma, 347, 170 – 184, https://doi.org/https://doi.org/10.1016/j.geoderma.2019.03.037, http://www.sciencedirect.com/science/article/pii/S0016706118319347, 2019.

Poggio, L. and Gimona, A.: Assimilation of optical and radar remote sensing data in 3D mapping of soil properties over large areas, Science of The Total Environment, 579, 1094 – 1110, https://doi.org/https://doi.org/10.1016/j.scitotenv.2016.11.078, http://www.sciencedirect.com/science/article/pii/S0048969716325177, 2017.

Ribeiro, E., Batjes, N., and Van Oostrum, A.: World Soil Information Service (WoSIS) - Towards the standardization and harmonization of world soil data. Procedures Manual 2018, Report ISRIC Report 2018/01, ISRIC - World Soil Information, http://dx.doi.org/10.17027/isric-wdcsoils.20180001, 2018.

---

## Author Response (AR2)

**SoilGrids 2.0: producing soil information for the globe with quantified spatial uncertainty**

**Response to reviewers**

Laura Poggio[1], Luis M. de Sousa[1], Niels H. Batjes[1], Gerard B. M. Heuvelink[1], Bas Kempen[1], Eloi Ribeiro[1], and David Rossiter[1]

[1]ISRIC - World Soil Information - Wageningen (NL)

Correspondence: Laura Poggio (laura.poggio@isric.org)

Reviewer comment:

Only one very minor comment may need to be considered by the authors.--Line 43: 'pedology-based' may be not an appropriate word here because modern digital soil mapping researches including the SoilGrids are also pedology-based. I suggest replacing it with 'polygon-based'. That is an obvious difference between traditonal and digital soil mapping. If agree, it would be better to make the change.

Response:

We modify the text as:

These issues have been addressed to varying degrees in various new global soil datasets (Batjes, 2016; Shangguan et al., 2014; Stoorvogel et al., 2017) that still largely draw on a traditional, pedology-based mapping approach (Dai et al., 2019)